Manuscript prepared for Atmos. Chem. Phys.
with version 2015/04/24 7.83 Copernicus papers of the LaTeX class copernicus.cls.
Date: 27 July 2016

# Measuring ice and liquid water properties in mixed-phase cloud layers at the Leipzig Cloudnet station

Johannes Bühl[1], Patric Seifert[1], Alexander Myagkov[1], and Albert Ansmann[1]

[1]Leibniz Institute for Tropospheric Research (TROPOS), Permoserstr. 15, 04318 Leipzig, Germany

*Correspondence to:* Johannes Bühl (buehl@tropos.de)

**Abstract.** An analysis of the Cloudnet dataset collected at Leipzig, Germany, with special focus on mixed-phase layered clouds is presented. We derive liquid and ice water content together with vertical motions of ice particles falling through cloud base. The ice mass flux is calculated by combining measurements of ice water content and particle *Doppler* velocity. The efficiency of heterogeneous ice formation and its impact on cloud lifetime is estimated for different cloud-top temperatures by relating the ice mass flux and the liquid water content at cloud top. Cloud radar measurements of polarization and *Doppler* velocity yield, that ice crystals formed in mixed-phase cloud layers with a geometrical thickness of less than 350 m are mostly pristine when they fall out of the cloud.

## 1   Introduction

Understanding the process of heterogeneous ice formation is currently one of the major topics in weather and climate research (Cantrell and Heymsfield, 2005; Hoose et al., 2008). Heterogeneous ice formation drives the generation of rain (Mülmenstädt et al., 2015), impacts cloud stability (Morrison et al., 2005) and atmospheric radiative transfer (Sun and Shine, 1994). It is therefore a crucial component in the hydrological cycle in the Earth's atmosphere. The interaction between aerosol and clouds in general involves very complex processes. Vertical motions keep mixed-phase clouds alive by activating aerosol particles to cloud droplets, while at the same time ice crystals nucleate and remove water from the cloud. To understand these complex interactions it is necessary to know all influences, process aspects, involved aerosol particles, cloud droplets, ice crystal ensembles as well as the spectrum of vertical air motions in detail. Laboratory measurements have already delivered a lot of useful information, e.g., about the ice nucleation efficiency of aerosol particles with temperature (Murray et al., 2012; DeMott et al., 2015). Observations of the process of ice nucleation in nature, however, are limited. By means of active remote sensing quantities that are directly connected with ice nucleation events, e.g., the ice-water content of ice crystals from cloud layers, can be measured (Zhang et al., 2010a; Bühl et al., 2013). In the European Union research project BACCHUS (Impact of Biogenic versus Anthropogenic emissions on Clouds and Climate: towards a Holistic

UnderStanding) the ice nucleating properties of aerosols are investigated. It is one major task of this project to study the life cycle of aerosols from its source through the clouds by means of aircraft, in-situ and remote sensing observations. Combined remote sensing observations in the framework of Cloudnet (Illingworth et al., 2007) constitute one main pillar of the BACCHUS project. Beyond other things, Cloudnet provides a target classification scheme for identifying the physical phase of hydrometeors. A similar multi-sensor approach is used by the ARM (Atmospheric Radiation Measurement) program (Shupe, 2007), which recently performed several measurement campaigns in the Arctic in order to study the interaction between aerosols and clouds (Zhang et al., 2014).

Since 2011, the Leipzig Aerosol and Cloud Remote Observations System (LACROS) (Wandinger, 2012) belongs to the Cloudnet consortium. In this article, remote measurements of LACROS analyzed with Cloudnet algorithms are used to describe ice formation processes under ambient conditions. Such remote sensing measurements fill a critical gap in the study of mixed-phase processes, because they deliver the information about the entire cloud column from the base to the top, which is not possible with aircraft measurements alone. In this way, the temperature level at which ice nucleation takes place can be derived and at the same time the resulting ice water falling from the layer can be analyzed.

Shallow mixed-phase cloud layers like altocumulus (Ac), altostratus (As) or stratocumulus (Sc) have been used before by different groups as atmospheric laboratories in order to study aerosol-cloud-dynamics interaction under ambient conditions (Fleishauer et al., 2002; Zhang et al., 2010b, a; Bühl et al., 2013; Schmidt et al., 2015; Seifert et al., 2015). These cloud types are especially well suited for process studies purposes, because they show narrow constraints on basic environmental variables like temperature, pressure, humidity and the number of potentially involved microphysical processes (Tao and Moncrieff, 2009). The well defined base and top of shallow cloud layers is optimum to study aerosol effects on ice nucleation as well as the impact of up- and downdraft on cloud ice production. As an additional benefit, these shallow cloud layers can easily be penetrated by lidar and cloud radar systems, which is not possible for deep convective clouds due to massive signal attenuation and strong turbulence within their cores. For climate research these shallow cloud layers are important due to their hard-to-predict impact on Earth's radiative budget. From the meteorological point of view, the understanding of ice formation processes in deep convective mixed-phase clouds may be more important. However, such clouds are difficult to observe and may not allow to resolve the basic ice processes and aerosol- and dynamics related aspects of ice formation. Both questions can be answered only by studying the process of ice formation itself in the atmosphere.

All of the statistical analysis of ice formation in our former studies (Kanitz et al., 2011; Bühl et al., 2013; Schmidt et al., 2015; Seifert et al., 2015), have been done manually. Such an approach is time consuming and cloud selection criteria can not be applied on a fully objective basis. Until now, some Cloudnet stations have been running continuously for more than 10 years (e.g., Chilbolton and Lindenberg), providing each day a wealth of measurement values. Therefore, the analysis of

clouds within such dataset can only be effective with an automated algorithm. For the present work, a method has been developed to automatically evaluate measurements from the Cloudnet dataset collected between 2011 and 2015 at TROPOS. A modified cloud-classification scheme from Bühl et al. (2013) is used to automatically discriminate liquid and mixed-phase cloud layers. The method is generally applicable to any Cloudnet dataset of arbitrary size. Hence, the method can be used to quickly analyze any dataset with the same objective criteria, and thus harmonizing Cloudnet measurements from all over the world.

The focus of the present work is twofold: Firstly, quantitative statistics about ice and water mass in shallow mixed-phase cloud layers are derived from the Cloudnet dataset, taking into account values of each Cloudnet profile individually. This constitutes a step forward compared to Bühl et al. (2013) where properties of ice and cloud water have been analyzed separately and independently. Secondly, statistics about *Doppler velocity (terminal fall velocity of the ice crystals plus vertical velocity of air)* and radar depolarization of the ice crystals are compiled in order to directly assess ice crystal sedimentation rates and to derive basic information about the shape of particles at the same time. (Not only quantitative knowledge about the particles themselves is gathered, but also the usability of cloud layers as atmospheric "laboratories" is characterized.) Only if ice crystals are pristine (i.e. do not show signs of riming growth, aggregation or secondary ice formation), there is a direct link between the properties of the ice (e.g., size, shape and mass) and their formation process within the mixed-phase cloud top layer. These measurements of ice particle properties are compared with laboratory studies of Fukuta and Takahashi (1999) in order to assess the quality of the Cloudnet measurements. Based on our dataset, the ice water content (IWC) produced by particles falling from cloud layers is derived and compared with the available liquid water within the cloud top layer. Together with the quality-assured measurements of fall velocity (Doppler velocity averaged over a complete cloud case) direct connection between the liquid water in the cloud top layer and the resulting ice mass flux is established, which can be regarded as a quantitative measure of heterogeneous ice formation in the atmosphere. With this approach, also the impact of ice formation on cloud lifetime is estimated for the temperature regime between $-35$ and $0\,°C$. Fukuta and Takahashi (1999) also provide comprehensive laboratory measurements of the growth of ice crystals. They found different distinct features in the resulting shape of ice crystals for different growth times and calculated corresponding residence times within a cloud layer, taking into account increasing fall speed with increasing particle size. For a residence time of 20 minutes within a mixed-phase cloud layer, particles could still be considered pristine. Also Yano and Phillips (2010) found that within this time, secondary processes like riming do not influence heterogeneous ice formation significantly. According to Fukuta and Takahashi (1999), a residence time of 20 minutes corresponds to a geometrical thickness of a mixed-phase cloud top layer of $350\,m$. Hence, for the present study only clouds with a geometrical thickness of below $350\,m$ are selected within this work to avoid altering of the ice crystals by riming, splintering, or aggregation processes.

The paper is structured as follows. Section 2 gives a short overview about the dataset used in the context of this work. In Section 3 the methodology to analyze the dataset is presented. At the beginning of Section 4 the ice-detection capability of different cloud radar systems is analyzed. After that, quantitative statistics of ice and liquid water within mixed-phase cloud layers are derived.

## 2 Dataset

The data analyzed within the frame of this work has been collected with LACROS (Wandinger, 2012) at TROPOS Leipzig, Germany (51.3° N, 12.4°E) between 2011 and 2015. The time coverage of Cloudnet observations at Leipzig is about $85\%$. Instruments relevant for the present work are the PollyXT Raman/depolarization lidar (Althausen et al., 2009; Engelmann et al., 2015), the Jenoptik ceilometer CHM15kx, the MIRA-35 cloud radar (Görsdorf et al., 2015) and the HATPRO (Humidity and Temperature Profiler) microwave radiometer (Rose et al., 2005). The measurements of these instruments are analyzed by the Cloudnet algorithms (Illingworth et al., 2007) to derive microphysical properties of hydrometeors on a continuous basis. Additionally model input of environmental variables like temperature and humidity is used. For the Cloudnet dataset of Leipzig, forecast data of COSMO-EU (Consortium for Small-scale Modeling - Europe) was used from 2011 to May 2014. Since June 2014, forecast data of the integrated forecast system of the ECMWF (European Centre for Medium-Range Weather Forecasts) was used. In the rare cases, when this data is not available, COSMO-EU is used as a fall-back option. The resulting Cloudnet dataset is the basis for the following analysis of cloud layers over Leipzig presented in the following.

## 3 Automated selection and classification of cloud layers in a Cloudnet dataset

The goal of this study is to obtain a dataset of mixed-phase cloud layers that fulfill certain quality criteria such as temporal and spatial homogeneity. As stated above, the continuous, homogenized Cloudnet-processed dataset is used as a basis for the approach. The automated Cloudnet algorithm reduces data from a set of remote sensing instruments on a common grid that has a temporal resolution of $30\,\mathrm{s}$ and a height resolution of $30.2$ m (similar to the one of the cloud radar). In a further step, the physical state of the atmosphere in all height bins is classified into different categories, e.g., containing cloud droplets, ice particles or both. Other definitions concerning aerosol are also present, but do not play a role in the context of this work. A detailed description of the target categorization scheme of Cloudnet is given in Illingworth et al. (2007). Basically, liquid water droplets are detected by a threshold in lidar signal followed by a characteristic decrease of the latter above liquid cloud base. Ice particles are in general defined to be present if the radar-observed vertical velocity of the targets indicates falling particles and the dewpoint temperature within a range gate is below $0\,°\mathrm{C}$. If, in addition, the analysis of the lidar signal of the considered pixel meets the criteria for the presence of liquid droplets, the pixel is categorized as mixed-phase. The height of the melting layer

is derived either from the meteorological data (dewpoint temperature is $0\,^\circ$C) or from measurements of radar linear depolarization ratio (LDR) larger than $-15\,$dB. Thus, the decision between liquid-only, mixed-phase or ice-only cloud layers is made primarily based on the modeled temperature and changes in the vertical-velocity profile. However, on the basis of temperature only, there is no way to unambiguously decide between drizzle and/or falling ice crystals below $0\,^\circ$C.

The target-classification of Cloudnet only takes into account single range-gates. Taking into account measurements of a complete cloud case facilitates the disambiguition between a mixed-phase and a liquid-only case. Hence, for this work, an automated algorithm has been developed that runs on this basic target-classification product of Cloudnet. Single 30-s profiles are analyzed to search for liquid water at $T < 0\,^\circ$C. If liquid water is found, the base and top height of the liquid layer is stored and the height-range below this liquid water bin is searched for ice. If ice is found below, also the height of transition between liquid and ice is stored. This procedure is done for all profiles of the dataset. Afterwards neighboring cloud profiles are merged to coherent cloud layers if they lie within $300\,$s of temporal and $350\,$m of vertical distance. The $300\,$s horizontal separation is derived from experience. Increasing the value increases the *homogeneity of the cloud cases, but reducing the the total number of cases at the same time*. The $350\,$m cloud thickness is motivated by Fukuta and Takahashi (1999), as it probably excludes secondary ice formation processes and particle riming. Cloud-top-height (CTH) of the cloud layers is specified to be larger than $1500\,$m in order to exclude clouds influenced by the boundary layer. Zhang et al. (2010a) went with a similar approach. A set of connected profiles constitutes a cloud layer for which we assume that the cloud properties are similar.

For the statistical analysis, a cloud must pass certain quality criteria: A coherent cloud structure must be found for more than 15 minutes, no seeding of particles from higher-level clouds must be present and at least $85\%$ of the cloud's occurrence time a liquid or mixed-phase cloud top must be detected (height-range where water vapor saturation over liquid water is close to 1, see Fig. 1). The properties of the detected clouds, e.g., cloud-top height (CTH), geometrical cloud thickness $\delta_h$, standard-deviation of cloud-top height $\sigma_{\mathrm{CTH}}$, cloud-top temperature (CTT), radar reflectivity factor (Z), ice-water content (IWC), liquid-water content (LWC), LDR, lidar attenuated backscatter coefficient ($\beta$) and lidar volume linear depolarization ratio are stored for further analysis. See Fig. 1 for an overview where the different properties are derived for one cloud case. The picture also shows, that some measurement values are taken only from a height-level $60\,$m below the mixed-phase cloud base. At this point, cloud droplets should be absent and ice particles should still be largely unaltered by evaporation or aggregation processes. Hence their size and shape should only be related to processes having taken place within the mixed-phase cloud top layer. In the context of this work, all measurement values derived in this way are marked with the index "CB" (for "cloud base"). *In addition, by this definition of coherent cloud layers the average vertical velocities can be used as an estimate of the particle fall velocity. In cloud layers, the size of turbulent eddies is restricted to the*

*layer depth. The maximum scale length of free turbulence within the layer is hence approximately two times its geometrical thickness (Moin, 2009). For cloud layers of $350\,m$ of vertical geometrical extent, such small scale fluctuations cancel out over the course of $15\,minutes$ or longer. The average Doppler velocity of falling particles measured over the timespan of cloud layer occurrence is therefore free of influences of small-scale turbulence. However, large-scale vertical air motions equal to or longer than $15\,minutes$ still influence the measurements.*

After cloud identification, the cloud-classification scheme from Bühl et al. (2013) is used to discriminate between liquid and mixed-phase cloud layers (see Fig. 2). This classification method reduces the dependence on model temperature by taking into account information from all cloud profiles to make a decision between the microphysical states "liquid" or "mixed-phase". Depolarization measurements from lidar and radar are used to directly identify ice crystals falling from a cloud layer. Mixed-phase clouds close to $0\,°C$ also often show a melting layer, which is the most unambiguous sign of the presence of ice particles (Di Girolamo et al., 2012). High LDR values are also produced by the needle-like ice crystals prevailing for clouds with a CTT between $-8$ and $-2\,°C$ (Fukuta and Takahashi, 1999). Such clear LDR signal make the decision between ice and liquid water fortunately very easy close to the $0\,°C$ level, where model temperature in most cases is not accurate enough and the increase in particle fall speed due to melting is not significant. For low values of Z (typically below $-30\,dBZ$) and no detection of a melting layer, the depolarized signal is usually too weak to be detected by the cross-polarized channel of the MIRA-35 cloud radar. In this case, measurements of volume linear depolarization ratio from a collocated PollyXT lidar is used (Engelmann et al., 2015), if available. In Fig. 3 three example cases with different CTT from different dates are shown together. Cloud radar measurements of $Z$, LDR and $v$ are shown together with the attenuated backscatter coefficient from the lidar. The CTT of the three cases are chosen in such a way that distinct differences in LDR measurements are visible between the cases. As an example for cloud detection/selection, all clouds with $\delta_h < 350\,m$ and $\sigma_{\mathrm{CTH}} < 150\,m$ detected on 2 October 2012 at Leipzig are marked in Fig. 4. The CTT statistics of all selected and classified cloud layers with these selection criteria ($\delta_h < 350\,m$ and $\sigma_{\mathrm{CTH}} < 150\,m$) are shown in Figs. 5a and 5b. It is visible that no mixed-phase clouds are detected below $-40\,°C$. *The result of this automated analysis is within the statistical accuracy of $15\%$ of the results of the study of Bühl et al. (2013) which was done on the basis on manual cloud selection.*

## 4 Quantitative description of heterogeneous ice formation in cloud layers over Leipzig

### 4.1 Ice-mass retrieval and detection thresholds

A quantitative retrieval of ice mass is done by Cloudnet via the method of Hogan et al. (2006). IWC values are obtained for each range bin with a simple empirical function depending on $Z$ and the ambient temperature. The uncertainty of the method is estimated by Hogan et al. (2006) to

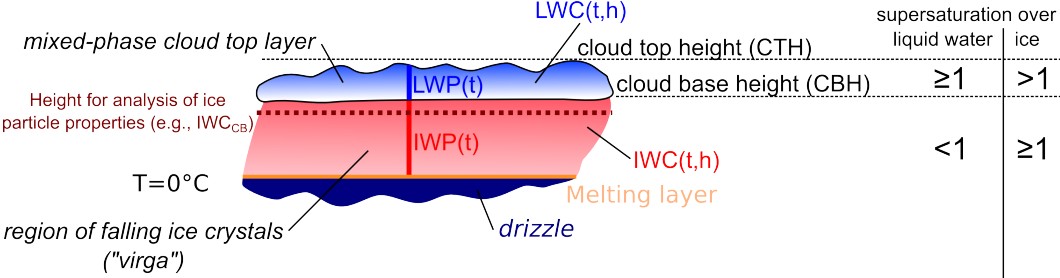

**Figure 1.** Schematic representation of the different measurement and averaging schemes in a mixed-phase cloud layer. Water droplets within the mixed-phase top layer are detected by lidar. The ice precipitation below is mainly detected by the cloud radar. IWC and LWC are provided by Cloudnet and are a function of height ($h$) and time ($t$). IWP and LWP are the column integrated values of LWC and IWC over the liquid cloud top and the ice precipitation, respectively. $IWC_{CB}$ represents the mean of all IWC values measured about $60\,m$ below current cloud base height (CBH). Following Zhang et al. (2014), state of water saturation is indicated for the different parts of the clouds.

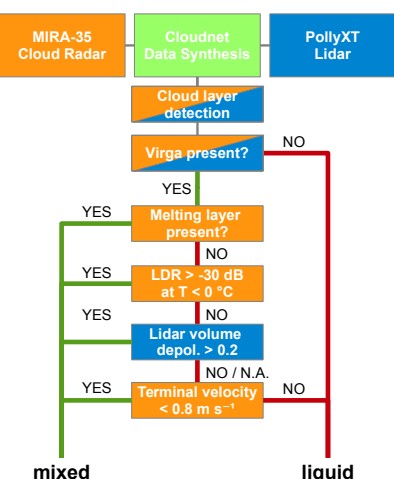

**Figure 2.** Flowchart of the mixed-phase cloud discrimination method from Bühl et al. (2013) as it is applied in the current work. Most clouds are successfully analyzed with combined lidar/radar.

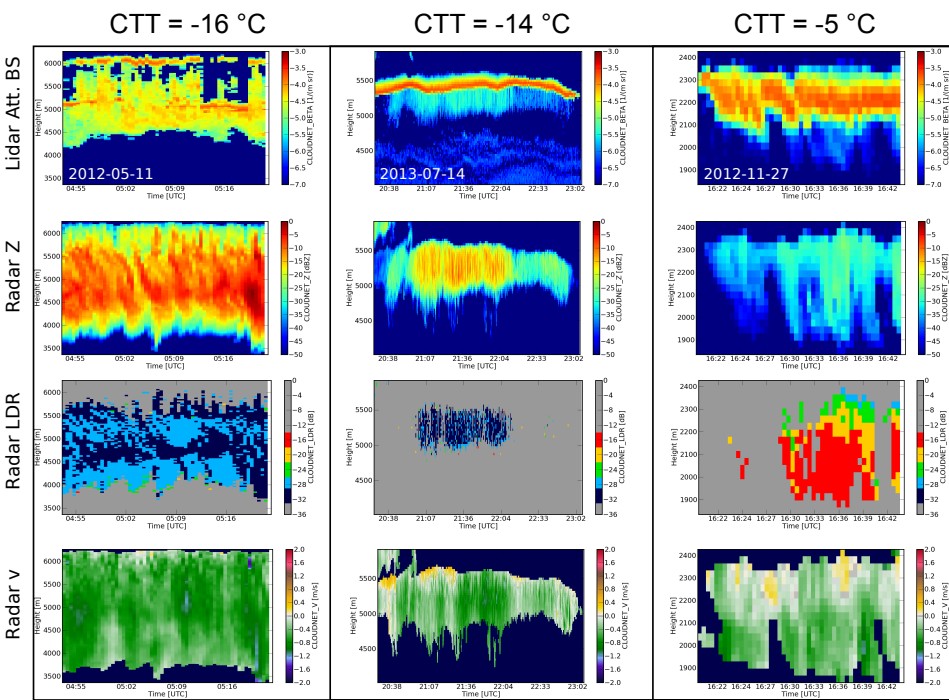

**Figure 3.** Three example case-studies of mixed-phase clouds identified with the automated algorithm described in Section 3.

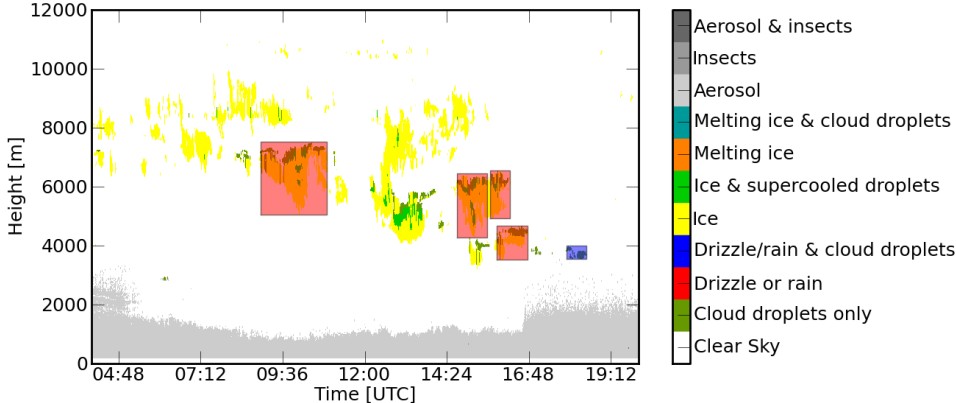

**Figure 4.** Example of automated detection of mixed-phase cloud layers on the basis of the Cloudnet target classification scheme for 2 October 2012. Clouds are marked due to the selection criteria explained in the text. Blue squares mark liquid-only layers and red squares mark mixed-phase layers. The colors are only for a very basic visualization of the layer detection. The decision between mixed-phase and liquid clouds in the following analysis is more complex and described in the text.

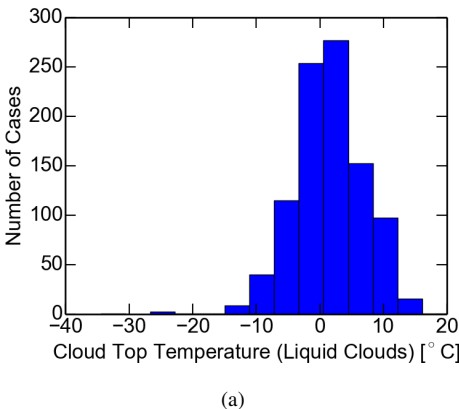 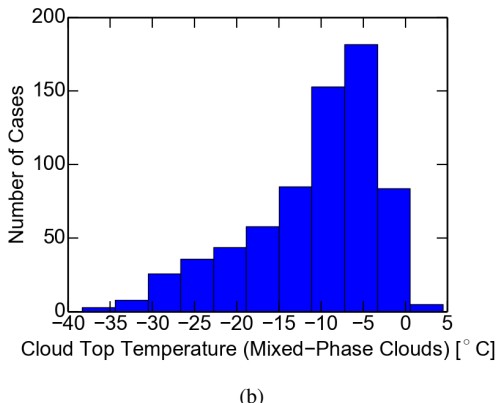

**Figure 5.** Distribution of cloud-top temperature for all pure liquid (a) and mixed phase (b) cloud layers detected between 2011 and 2015 over Leipzig.

be $(+50/-30)\%$ below a temperature of $-10\,°C$ and $(+100/-50)\%$ above. A possible bias of $(+15/-10)\%$ is estimated by Hogan et al. (2006). *Comparison between the retrieval of Hogan et al. (2006) and three other Z-T parameterizations from Protat et al. (2007) shows that the spread between the methods is about a factor of two for the temperature interval -30 to -15°C and about a factor of 5 in the interval -15 to 0°C. Other methods to derive the IWC from continuous remote-sensing observations, as they are for instance summarized in Shupe et al. (2008), suffer from their restrictions to certain scenarios or they don't take into account any temperature dependence of the ice properties.*

Uncertainties in the measurements of $Z$ add to these errors. Amongst these, for the quantitative understanding of ice formation in the atmosphere, knowledge about the accuracy and – especially – about the signal detection threshold of the cloud radar is critical. In the case of ground-based radar, different factors can affect the measured values of $Z$, e.g., unknown attenuation in rain and uncertainties in radar calibration. Attenuation induced by water vapor and liquid cloud layers is corrected in Cloudnet. Additionally, attenuation is avoided by excluding clouds from the analysis that are measured above other clouds or rain. *The LACROS cloud radar is calibrated by the manufacturer with the method described in (Görsdorf et al., 2015).* The calibration is estimated to be accurate to $3\,dB$, resulting in an additional bias in the IWC retrieval of about $35\%$ (for the range between $-60$ and $0\,dBZ$ and $-40$ to $0\,\circ C$), making them an estimation within the order of magnitude.

The starting point for the characterization of the IWC dataset is Fig. 6. In this figure, the signal-to-noise ratio (SNR) detected within cloud virgae (streams of ice particles falling from cloud top in which water is close to saturation over ice, see Fig.1) is depicted together with the detected average LDR (color scale). The LACROS cloud radar can detect a signal down to a SNR of $-23\,dB$. From Fig. 6 it becomes obvious that particle detection at higher temperatures above $-10\,°C$ are often close to the detection limit. In this temperature regime, the detection of some ice below cloud bases might

be missed and clouds could be erroneously be classified as liquid-clouds. In contrast, ice detection seems to be quite reliable below $-10\,°C$, where all cases have a mean SNR well above the detection threshold. It is also visible from the figure, that LDR values can only be detected if a certain SNR threshold is reached.

Figure 7a depicts all measurements of $Z_{CB}$ sorted by CTT. In Fig. 7b the values of $Z_{CB}$ are shown averaged for individual cloud cases. The equivalent values of $IWC_{CB}$ are shown in Fig. 7c. The LACROS MIRA-35 cloud radar has a detection threshold of $Z_{thr} = -45\,dBZ$ at a range of $5000.0\,m$ (Görsdorf et al., 2015). For other ranges $r$ we hence find a threshold of

$$Z_{thr} = -10 \times \left(2\log(5000^2/r^2\right) - 45\,dBZ, \tag{1}$$

due to the quadratic decrease of received radiation with range. The corresponding thresholds of IWC ($IWC_{thr}$) for different radar systems are drawn within the plots. Please note that the ice detection threshold is not only depending on the radar signal threshold, but also on temperature, according to the retrieval of Hogan et al. (2006). For spaceborne systems $Z_{thr}$ is nearly constant for the complete troposphere. The measurement distance of about $400 - 800\,km$ leads to a range-induced signal variation of maximum $5\%$ between $0$ and $12\,km$ height. For ground-based systems, however, the detection threshold varies significantly for different heights. This phenomenon is depicted in Fig. 7d, where mean $Z_{CB}$ is plotted against CTH instead of CTT. The height-dependent detection threshold of the LACROS cloud radar is shown.

The LACROS cloud radar has a depolarization decoupling of $-33\,dB$, which stands out from all radars currently operated within the framework of Cloudnet. Only this technical prerequisite makes high-quality measurements of LDR possible. Also the detection threshold of $-47\,dBZ$ at a range of $5000\,m$ is outstanding. Satellite missions equipped with cloud radars like Cloudsat (Stephens et al., 2002) and EarthCare (Illingworth et al., 2014) have detection thresholds within the troposphere of $-27\,dBZ$ and $-33\,dBZ$, respectively. Hence, the CloudSat and EarthCare satallites are both able to detect most of the ice formation in clouds with CTT $< -10\,°C$. At temperatures warmer than this, probably $90\%$ of the ice-signals below the cloud layers will be missed (see Fig. 7a).

### 4.2 *Particle Doppler* velocity and radar depolarization of pristine ice crystals

In contrast to the extensive properties $Z_{CB}$ and $IWC_{CB}$, the measurements of the cloud radar can also be used to derive the intensive properties of the ice crystals (e.g., $v_{CB}$ and LDR). The latter are connected to size, shape and orientation of the ice particles. Values of LDR and $v_{CB}$ averaged for each cloud case are shown in Figs. 8c and 8d. Note that LDR is dependent both on particle shape and particle orientation, so this information is not unambiguous (Reinking et al., 1997). However, if particles are oriented, high LDR values indicate prolate (column-shaped) particles and low values point towards more oblate particles like dendrites. For randomly oriented aspherical particles, LDR

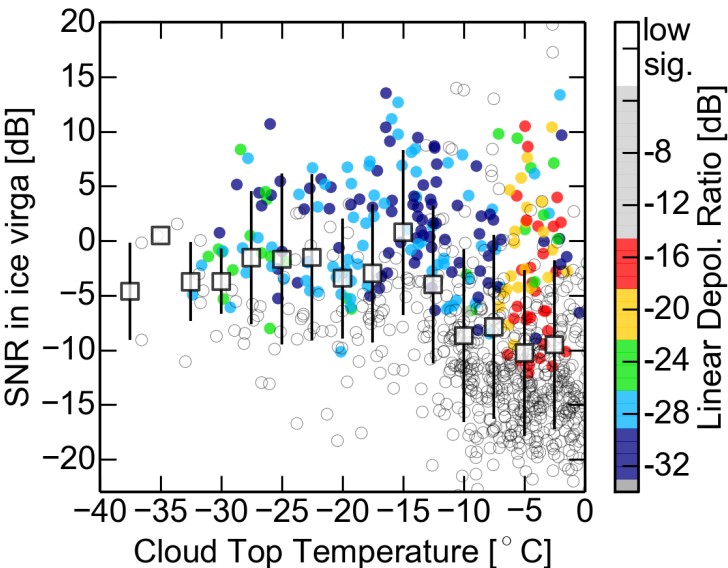

**Figure 6.** The 90% percentile of cloud-radar SNR is shown for each cloud case together with mean detected LDR. For $\pm 2.5\,^{\circ}$C intervals mean values (white squares) and standard deviation (black bars) are given.

is always elevated. In this way, LDR gives only basic information about particle shape, but LDR has the advantage that it can be derived easily together with $v_{CB}$ values with a vertical pointing radar.

The single values (30-s integration time and 30 m height resolution) of LDR and $v_{CB}$ from all cases are shown in Figs. 8a and 8b. The values are taken from the virgae where the target classification of Cloudnet states "ice only" (red-zone in Fig. 1). These representations already show interesting features. In Fig. 7a, e.g., it has already been shown that at temperatures above $-10\,^{\circ}$C the average value of $Z_{CB}$ is often below $-30$ dBZ. The depolarization measurements show a clear feature of elevated LDR values in this temperature range, pointing towards the presence of highly prolate and oriented ice particles. The vertical velocity measurements in 8b also show features of enhanced Doppler velocities indicating the different prevailing particle habits over the temperature range of heterogeneous ice formation.

Fukuta and Takahashi (1999) also found several distinct features in the distribution of ice particle size, shape and mass with temperature. Some of these features can be seen within the measurements of LDR and $v_{CB}$:

– An enhanced growth of ice crystal mass around $-14\,^{\circ}$C was found by Fukuta and Takahashi (1999). The effect can also be seen in Figs. 7a and Fig. 7b as a strong increase of $Z_{CB}$ at this temperature.

– The high values of LDR measured at a CTT of $-5\,^{\circ}$C correspond to a needle- or column-like particle shape (see Figs. 8a and 8c). In the temperature range around $-14\,^{\circ}$C LDR values can be found to be around $-28$ dB, corresponding to plate-like crystal shapes. Please note that

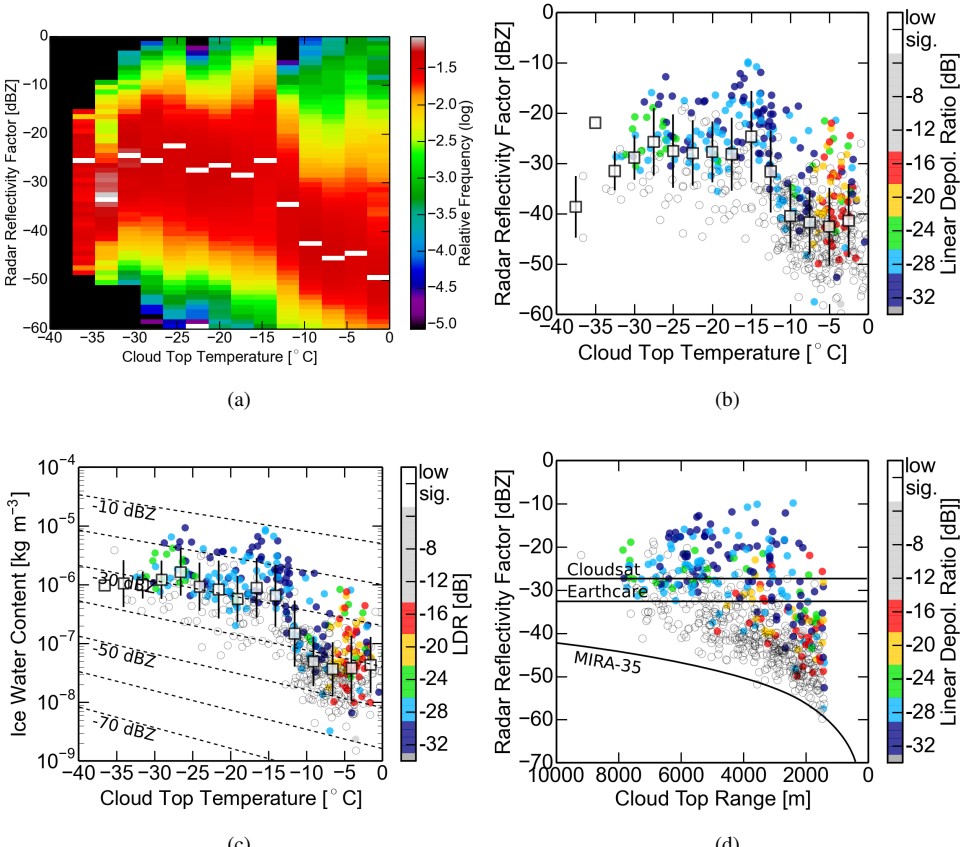

**Figure 7.** (a) All values of $Z_{CB}$ column-normalized. Maximum values in each column are marked with white bars. (b) $Z_{CB}$ averaged for each cloud case together with averaged LDR values. (c) $IWC_{CB}$ averaged for each cloud case. (d) values of $Z_{CB}$ depicted depending on CTH instead of CTT; the cut-off at lower heights appears due to the selection criterion CTH $> 1500$ m. Thresholds for $Z$ and IWC are illustrated within the graphs as solid lines with labels.

these features are also displayed in Fig. 3. In Reinking et al. (1997) the LDR values values of
$-15$ to $-20$ dB are computed for these ice crystals shapes.

– Hints on the presence of these isometric ice crystals are found in the increase of Doppler velocity in Fig. 8d. Measured Doppler velocities peak at around $-10$ and $-22\,°$C, while minima of LDR can be found at $-12$ and $-22\,°$C. This connection also points towards more isometric, more compact ice crystals around these temperatures. Actually, the increase of Doppler velocities in the temperature interval between $-5$ and $-0\,°$C bin is also found in (Fukuta and Takahashi, 1999). However, the uncertainty of the measurements in this temperature interval is too large for a definite identification of this phenomenon.

*We note that by our definition of "pristine particles" we follow the laboratory experiments of Fukuta and Takahashi (1999). We consider all particles pristine that have not undergone riming*

*growth, aggregation or splintering. As explained above, these processes should be excluded by the*
*cloud selection criteria. Non-pristine crystals that would result, e.g., from ice particle break up, ag-*
*gregation or graupel formation would be asymmetric and would therefore increase the LDR values.*
*However, the LDR values we find are very close to the literature values of Reinking et al. (1997),*
*which yield* $-28\,dB$ *for plate-like crystals and about* $-20\,dB$ *for columnar shaped particles. These*
300 *calculations however depend strongly on the orientation of the ice crystals (i.e. at what angle they*
*"wobble"). Larger angles of orientation would increase the measured LDR values. Finding these*
*very low values of down to* $-30\,dB$ *is therefore an indication that pristine particles are dominating*
*and secondary ice formation only plays a minor role within the selected cloud layer type.*

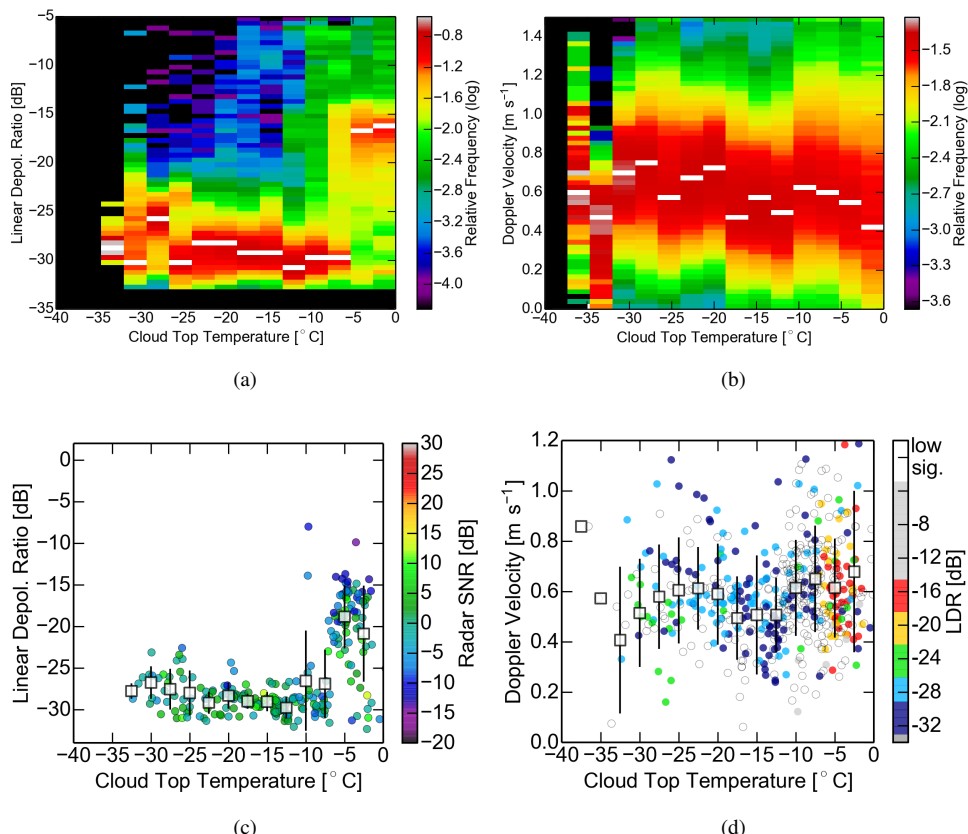

**Figure 8.** All values of (a) LDR and (b) $v_{CB}$ measured with cloud radar MIRA-35 in the virgae below cloud layers over Leipzig. The visible spread in $v_{CB}$ is due to vertical air motion (see velocity plots in Fig. 3). Averaged values for the individual cloud cases are depicted in (c) and (d), respectively. Maximum values in each column are marked with white bars.

### 4.3 IWC$_{CB}$ and LWC at cloud top

In the previous sections the properties of the ice particles produced within mixed-phase clouds were investigated. For the estimation of cloud stability by approaches like the one presented by Korolev

and Field (2008), however, the ratio of $IWC_{CB}$/LWC=ILCR (ice- to liquid water content mass ratio) at cloud top is important. For that estimation, the LWC has to be retrieved in addition to the IWC.

In this work, the liquid-water content (LWC) of a cloud layer is calculated for each cloud pro-
file adiabatically between cloud bases and cloud tops, assuming an adiabaticity of 1. Cloudnet also provides operationally adiabatic profiles scaled with the LWP measured with the microwave ra-
diometer (Merk et al., 2016). However, the LWP measurements of the microwave radiometer have an uncertainty of about $\pm 20\,\mathrm{g\,m^{-2}}$. Since the average liquid water path of the cloud under study is actually around $20\,\mathrm{g\,m^{-2}}$, the adiabatically calculated profiles are used in the context of this work. An overview about the LWP of all cloud layers under study is given in Fig. 9a. Zhang et al. (2014) found a similar relationship between LWP and T for Arctic supercooled mid-level clouds. For the current work, the retrieved adiabatic LWP can be considered as a maximum guess. The actual LWP may be lower, which is described by the adiabaticity factor $f$. Merk et al. (2016) report $f$ to be within 0.6 to 1.0. Airborne studies of mixed-phase clouds found rather good agreement between observed and adiabatic LWC profiles for shallow cloud layers (Larson et al., 2006; Noh et al., 2013). Hence, the adiabatic LWC profiles serves as an estimation until better calibration methods for the microwave radiometers are available. Such methods are currently under investigation by different groups, e.g. Maschwitz et al. (2013). An *alternative* approach may be lwp measurements with depolarization lidar (Hu et al., 2010; Donovan et al., 2015).

In Fig. 9b, $IWC_{CB}$ is divided by the mean LWC in the mixed-phase cloud top in order to derive an estimate of ILCR. Assuming that particles directly below the mixed-phase layer have the same properties as within the layer, this estimate of ILCR is representative for the average ratio between ice and liquid water content within the mixed-phase cloud layer. The uncertainty of ILCR is still quite large. *As mentioned above, the absolute accuracy of the measurements of LWC and $IWC_{CB}$ is one order of magnitude due to unknown biases of the retrieval itself and the radar calibration. Nevertheless, the standard-deviation within a temperature interval of about $-5\,^{\circ}C$ is only a factor of* 2. That comparably low value might be partially due to the reason that both the $IWC_{CB}$ and the LWC retrieval method rely on the same temperature field, reducing this part of the *variability*. Systematic uncertainties of both the $IWC_{CB}$ and LWC, however, remain.

## 4.4 Estimating the ice mass flux from a cloud layer

The ILCR connects measurements of ice and liquid water mass. However, ice crystals formed inside the mixed-phase cloud top layer are falling with $v_{CB} > 0.2\,\mathrm{m\,s^{-1}}$ (see Fig. 8d), while the majority of cloud droplets have negligible fall velocities. The same number of particles creates a different IWC when falling at different terminal velocities, because the stream of particles is "stretched" differently. Hence, the ice flux $F = IWC_{CB} \times v_{CB}$ at cloud base gives the most accurate description of ice formation per time interval inside the cloud top layer. In this very simple picture, $F$ describes the flux quite coarsely. However, since both $v_{CB}$ and $IWC_{CB}$ are calculated from the same radar

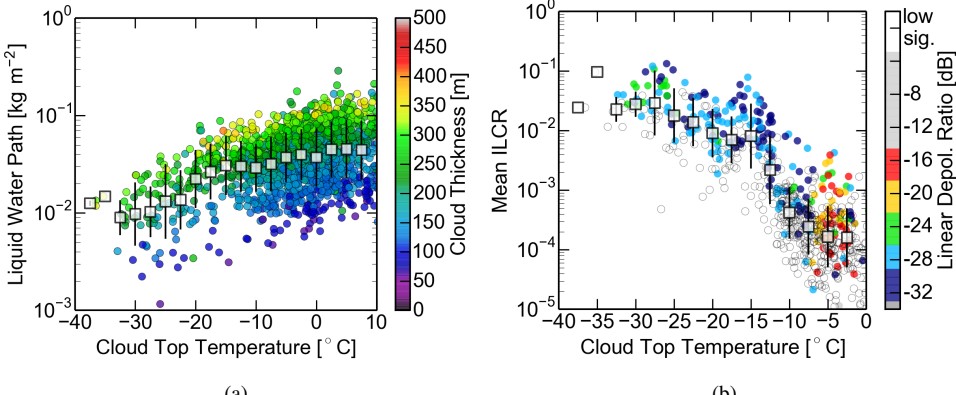

**Figure 9.** (a) LWP of all clouds under study is shown in dependence of temperature and mean cloud top thickness. (b) The ratio between $IWC_{CB}$ and mean LWC is calculated for each cloud-profile and averaged for each cloud case.

signal, a direct multiplication can be applied. The resulting parameter is an estimation within the order of magnitude, but it can savely be compared to the other flux values, presented here. Figure 10a

345 displays averaged $F$ for all cloud cases under study. Especially at temperatures below $-20\,°C$ it can be seen that the flux of ice mass is only weakly depending on temperature. In this temperature range $IWC_{CB}$ (Fig. 7c) is decreasing with temperature while $v_{CB}$ (Fig. 8d) is increasing. Also the peak at $-15\,°C$ is less pronounced compared to Figs. 7b and 7c as it coincides with a minimum in particle fall velocity.

The concept of ice mass flux also opens the possibility to derive basic information about the impact of ice formation on static cloud lifetime. Water particles most probably glaciate at cloud top and fall through the mixed-phase layer. Having connected $v_{CB}$ with with $IWC_{CB}$ to the ice flux, it is also possible to relate this quantity to the available LWP within the ice-generating liquid cloud layer. Since ice particles grow through the Wegener-Bergeron-Findeisen process (Korolev and Field,

2008), there is an indirect connection between the amount of available water vapor and ice crystal growth. Hence, a dynamic view of ice formation in the cloud layers can be established by dividing $F$ and LWP profile-wise.

$$\frac{LWP}{F} = \frac{LWP}{IWC_{CB} \times v_{CB}} \stackrel{!}{=} T_l, \qquad (2)$$

Defined in this way, $T_l$ is a time measured in seconds. Assuming *static* conditions $T_l$ is the time

the liquid cloud top layer would have depleted all its liquid water by ice sedimentation alone. It is a theoretical quantity, but it gives an impression of the relative impact of ice formation on different cloud layers. An overview of $T_l$ for all cloud cases under study is shown in Figure 10b, indicating that $T_l$ varies over 4 orders over the temperature range of heterogeneous ice formation ($-40$ to $0\,°C$).

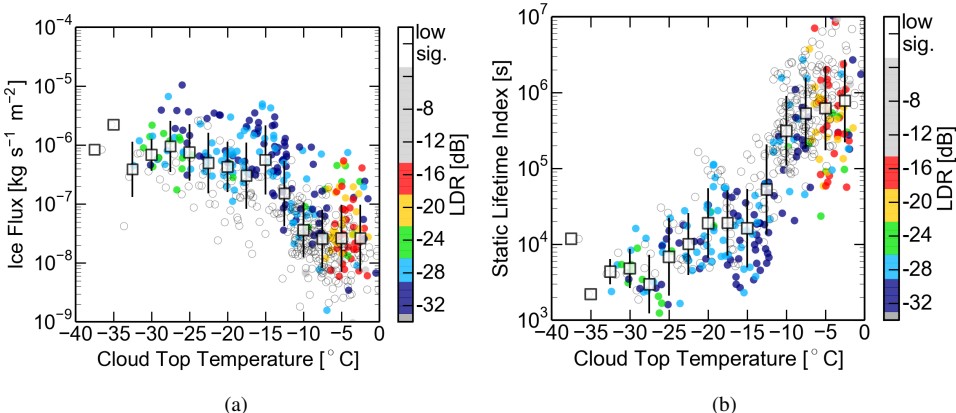

**Figure 10.** (a) The ice mass flux at cloud base. (b) The estimated static lifetime index $T_l = \text{LWP}/F$ of each cloud.

## 5 Summary and Conclusions

Quantitative retrievals of ice crystal properties like basic information about particle shape and *Doppler* velocity have been found to be quantitatively in line with theoretical computations of Reinking et al. (1997) and laboratory studies of Fukuta and Takahashi (1999). The dominating part of the ice particles falling from mixed-phase cloud layers with a geometrical thickness of the mixed-phase top layer $< 350\,\text{m}$ are apparently mostly pristine. Hence, these particles are probably the result of primary ice formation and secondary ice formation is a minor process in these cloud layers. Additionally, a profile-based connection between the measured liquid water path (LWP) and the retrieved $\text{IWC}_{\text{CB}}$ has been established. The flux of ice mass at cloud base height is found to increase within two orders of magnitude within the CTT range from $-40$ to $0\,°\text{C}$. The relative influence of the loss of ice on static lifetime index is found to increase even by $4$ orders of magnitude within the same range of CTT.

It is demonstrated in this work that a detailed insight into the microphysics of mixed-phase cloud layers is possible with a combination of the LACROS instrumentation and Cloudnet. Vertical velocity measurements show the dynamical state of the turbulent layer and cloud radar measurements show the ice flux from that layer. Together with the retrieval of ice nuclei properties with Raman lidar (Mamouri and Ansmann, 2015) the life cycle of an ice nucleus in mixed-phase clouds from entrainment over activation to ice nucleation and sedimentation can be closed.

It is an important finding that the dominating part of ice crystals produced mixed-phase cloud layers with $\delta_h < 350\,\text{m}$ are pristine. This means that the flux of ice crystals measured at cloud base is directly connected to the rate of ice nucleation within the mixed-phase layer. The direct measurement of the complete process of ice nucleation seems therefore feasible with remote sensing. However,

in future, more advanced particle typing methods such as presented in Myagkov et al. (2015a, b) should be applied to further characterize shape and size of the particles on an operational basis.

The relative impact of the loss of ice water on a mixed-phase cloud layer can be measured. However, it has to be noted again, that the cloud *static lifetime index* presented here might not directly
be connected to the absolute lifetime of a cloud. Even the definition of a cloud lifetime is difficult, because particles are mixed between cloud parcels and the apparent motion of clouds can be independent from horizontal wind speed. However, the static lifetime value presented here can be used to study the impact of ice on predominantly liquid cloud layers occurring at different temperature levels. Measurements of ice mass flux and the static lifetime index $T_l$ indicate a minimum cloud
layer lifetime of $3$ hours at $-25\,^{\circ}\text{C}$ (see Fig. 10b). At temperatures above $-15\,^{\circ}\text{C}$ the relative impact of ice formation has already shrunk by 2 orders of magnitude. Given the fact that Korolev and Field (2008) showed that the cloud layers under study here actually are able to recreate liquid water via recurring upward air motion, these clouds seem to be extremely stable with respect to water depletion due to ice formation. The static lifetime index is a step forward compared to Bühl et al. (2013),
where the mass ratio of ice and liquid water in mixed-phase layered clouds was estimated with a ratio of IWP and LWP on manually selected clouds. The ratio of $\text{IWC}_{\text{CB}}$ and LWP, combined with the particle Doppler velocity gives a much more direct measure of the actual impact of the ice on the liquid water within a mixed-phase layer.

The presented algorithm to classify mixed-phase clouds in Cloudnet datasets is universal. It is
405 not only applicable on Cloudnet datasets, but in general on all datasets that separate an atmospheric column into liquid, ice and mixed-phase. The evaluation of mixed-phase clouds predicted by weather models seems therefore possible if suitable data output is given.

*Acknowledgements.* The research leading to these results has received funding from the European Union Seventh Framework Programme (FP7/2007-2013) under grant agreement numbers 262254 (ACTRIS) and 603445
(BACCHUS) and from the HD(CP)$^2$ project (FKZ 01LK1209C and 01LK1212C) of the German Ministry for Education and Research.

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
