# Peer review of "Measuring ice and liquid water properties in mixed-phase cloud layers at the Leipzig Cloudnet station"

_Atmospheric Chemistry and Physics, 2016_

## Referee Comment (RC1) · Anonymous Referee #1 · 24 Feb 2016

Overall summary: This manuscript used measurements collected by Leipzig Aerosol and Cloud Remote Observations System (LACROS), which includes Raman lidar, ceilometer, cloud radar and microwave radiometer, and then were analyzed with Cloudnet algorithms to take a detailed insight into the microphysics of mixed-phase cloud layers. Authors found that shallow mixed-phase cloud layers mainly produce pristine ice and spaceborne cloud radar might miss a large part of ice formation. This work presents valuable information to understanding of ice formation and to accuracy of satellite measurements. Some minor questions/suggestions need to be solved are listed in the following: Comment and Question: 1. Line 64, 97 and 115: Authors should define the acronyms (TROPOS, LDR, COSMO-EU) when it firstly appeared in the article. 2. A suggestion: the paragraph 2 in page 2 (Line 39-line56) is better moved to the ending of the introduction. 3. As we know, multilayered cloud systems very fre-

quently occur in the atmosphere (Huang, J., P. Minnis, B. Lin, Y. Yi, S. Sun-Mack, T. Fan, and J. Ayers, Determination of ice water path in ice-over-water cloud systems using combined MODIS and AMSR-E measurements, Geophysical Research Letters, 33 (21) (2006), L21801, doi:10.1029/2006GL027038 ; Huang, J., P. Minnis, B. Lin, Y. Yi, M. Khaiyer, R. Arduini, A. Fan, and G. Mace, Advanced retrievals of multilayered cloud properties using multispectral measurements, Journal of Geophysical Research, 110 (D15) (2005), D15S18, doi:10.1029/2004JD005101.). For this study, all lidar or Radar profiles were used? How did you considered the attenuation of lidar power when signal penetrate the lower cloud layer in the multilayered cloud system? 4. For figure1: 'On top the predominantly liquid water top is detected by lidar'. As we know that lidar signal is hard to penetrate mix-phase cloud layer, how could it detects the liquid water top? And the schemes of mixed-phase cloud layer are not well described. If there is mixed-phase cloud, why IWP is only below cloud layer? 5. Line 325: 'a minimum cloud layer lifetime of 3 hours around -25°C', how could authors get this value herein. 6. For the retrieval of LWC, the adiabatic model was used in this study. However, the entrainment of cloud top should be considered, thus this process may reduce the LWC. A better method possible is based on the depolarization ratio measurement from lidar (See paper: Hu, Y., S. Rodier, K. Xu, W. Sun, J. Huang, B. Lin, P. Zhai, and D. Josset, Occurrence, liquid water content, and fraction of supercooled water clouds from combined CALIOP/IIR/MODIS measurements, Journal of Geophysical Research, 115 (2010), D00H34, doi:10.1029/2009JD012384. Hu, Y., M. Vaughan, C. McClain, M. Behrenfeld, S. Sun-Mack, D. Flittner, J. Huang, B. Wielicki, P. Minnis, C. Trepte, and R. Kuehn, 2007: Global statistics of liquid water content and effective number density of water clouds over ocean derived from combined CALIPSO and MODIS measurements, Atmos. Chem. Phys., 7, 4065-4083.)

---

## Referee Comment (RC2) · Anonymous Referee #2 · 22 Apr 2016

The authors of the manuscript used the Cloudnet algorithm to analyze mixed phase clouds observed in Leipzig. I find the topic interesting and the article is in general worth to be published. However, I have some major comments which should be addressed. In particular, the authors have to address uncertainties more carefully and summarize their findings better so that they not oversell their results.

**1  Major comments**

Abtract: In the abstract, the sensitivity of space borne radars is discussed. However, this issue is mentioned only in Figure 7 and the last(!) sentence of the summary. When mentioning this in the abstract, the reader expects a much deeper discussion of that

issue. How is this topic related to the key questions of the paper? I think the authors should either discuss that topic in greater detail in the paper or remove it from the abstract.

Section 4.1: My major concern is that Cloudnet's IWC estimations are used uncritically. Even though an uncertainty estimate is presented in L 167, the IWC results are likely biased and the authors should make that clear. How does that impact the authors' results and conclusions? What happens if other Z-IWC relations are used? From the spread of results when using other Z-IWC relations, is it possible to say something about the robustness of the key results of the study? Moreover, the authors should discus in more detail why is there this drop around -10°C? How can the authors distinguish between impact of particle type and number concentration? Can the authors exclude the possibility of cloud misclassification (i.e. the clouds are actually liquid)? Why is there no significant decrease for very cold temperatures when I would expect smaller particles as well?

Section 4.2: I cannot see that Figure 8 shows the 'necessity to select thin clouds'. I see that IWC is higher for lower temperatures, but how do the authors now that this is due to more non-pristine particles? I would actually recommend to omit Figure 8 (or explain better why its interesting), because the authors motivated the use of a maximum thickness of 350 m already well with references to other studies.

Section 4.3: What is the influence of vertical air motion? Did the authors correct for that?

Section 4.3: One of the key conclusions of the authors is that 'ice crystals formed in cloud layers with a geometrical thickness of less than 350m are mostly pristine when they fall out of the cloud' which they support by agreement of observations to literature studies. I think this is not supported by the study (except maybe the 0 to -10°C range). Even though they show indications for the presence of pristine particles, but they show not evidence that other particles are not present. How would non-pristine particles

appear in the data with respect to LDR and fall velocity?

Section 4.4: How do the authors know that the median of LWP was around 20 gm-2? From the radiometer of from the adiabatic profiles? Are there differences in the statistic? Even though the general uncertainty of LWP retrievals is around 20 gm-2, the uncertainty is actually way less if environmental conditions (mainly water vapor) are known. As far as I know, LWP estimation is not standardized in Cloudnet. But if LWP is bias corrected before the cloud (i.e constrain the water vapor), I expect the radiometer to be a better source for LWP than the adiabatic profiles. The adiabatic profiles are more an upper threshold for LWP. Moreover, your analysis of ILCR depends on uncertainties and biases of both LWC and IWC.

Section 4.5: As the authors note by themselves, this analysis is only for static conditions. I would propose that this can be seen from the nomenclature as well. I would suggest to call the lifetime index e.g. static lifetime index or potential lifetime index etc.

Summary:

L 303-305: I guess the authors refer to Figure 10? This is interpretation should have been discussed earlier in the manuscript. Moreover, it does not hold given the uncertainties of both LWP and IWC and that the authors show no quantitative evidence that IWC can be really estimated from LWP and CTT.

L 315: See comment to section 4.3

L 340-342: See above, I did not find any relation established and evaluated in this manuscript.

L 346-349: See also comment to abstract: In general, authors should avoid raising totally new issues in the summary as it is done here. Maybe, this part can be moved to the discussion of Figure 7? Is that actually a new, relevant result that pristine ice crystals have low backscattering cross sections and might be missed if the radar sensitivity is not sufficient?

**2 Minor comments**

Title: I find the title of the article misleading. Even though a ratio between LWC and IWC is introduced, it is not discussed in detail.

Figure 6 and other: If the authors plot the lines on top of the dots, the figure would be much easier to read.

Figure 7a and other: What is indicated by the white boxes?

L. 93: Add Hatpro Reference

L. 98: Because the analysis depends on model temperature, what is the impact of this change on your analysis? Is there a bias between the models?

L. 118: add 'in Cloudnet below $0°C$'

L. 120: I found this part hard to understand. Please motivate more clearly why you need this algorithm on top of Cloudnet and what is does

L. 124-128: What is the motivation for these thresholds?

L. 125: What is meant by 'driving microphysics'?

L. 139: I found it puzzling that CB refers to the cloud base of the mixed phase cloud and not of the complete cloud. What is actually the author's definition of a cloud? Are pristine ice particles already precipitation and not considered a cloud anymore? The same holds for Figure 2: 'signal below cloud layer' makes little sense if it is not clear that the mixed cloud layer is meant.

L. 151: At a typical radar range, what Ze value does -10 dB SNR correspond to?

L. 160: Please motivate thresholds already here.

L. 187: Is this equation valid for all radars? Where is IWC_thr in the plot?

L. 192: Replace dramatically by e.g. significantly

L. 194: The threshold is black instead of red. In general, such descriptions should be indicated in the caption instead of the text.

Figure 1: My printer did not print the red ice part of the cloud which makes the figure very hard to understand.

Figure 4: Mention why other clouds are not considered.

L 222: Replace 'one has to keep in mind' with 'note'

L 224: Are the particles investigated here within this interval?

L 250: Isn't the fall velocity increased because these particles are more compact?

L 273: ')' missing

L 275: Remove actually

L 278: too colloquial: 'actually makes sense'

L 302: replace 'seem to be' with 'apparently'. 'seems to be' indicates something that is not true in reality.
* * *

---

## Referee Comment (RC3) · Anonymous Referee #3 · 28 Apr 2016

The authors present an analysis of a long-term dataset of mid-latitude mixed-phase cloud properties observed by a combination of Ka-band cloud radar, lidar and microwave radiometer. The Cloudnet categorization is used in combination with additional algorithms to estimate liquid and ice water content and ice particle motion to derive ice mass flux.

In general, I find the study interesting and certainly worth for publication in ACP. Unfortunately, the manuscript does not adequately discuss previous work in this field. I also had the feeling that many statements and conclusions given miss proper discussion. I therefore recommend publication after the comments and corrections listed below are addressed.

General comments:

[Figure]

Lack of references and discussion of previous work:This manuscript puts a "special focus on mixed-phase cloud layers" and aims at characterizing heterogeneous ice formation within them with a combination of ground-based remote sensors. However, while reading through the manuscript, I found the given references and discussion of the results of this study with former work to be rather insufficient. Mixed-phase clouds either in the arctic or in mid-latitudes have been a focus topic during the last decade of many institutions and observing programs (for example the ARM program). Nevertheless, I hardly could find any citation of the important work which has been done in this field in this manuscript. I especially miss a proper citation in the introduction but also in the discussion of the results. In my opinion, a proper discussion of the work which has done in this field would also help to put your results into perspective and would actually strengthen your study. Just to give one example: I did just a half an hour literature review on this topic and found for example a very similar comparison between LWP and cloud top temperature as you show it in Fig. 10a in a recent paper by Zhang et al., "Ice Concentration Retrieval in Stratiform Mixed-Phase Clouds Using Cloud Radar Reflectivity Measurements and 1D Ice Growth Model Simulations", JAS, 2014 (their Fig. 7). Although their paper is about arctic mixed-phase clouds, such work should be discussed in your paper. In fact, I could imagine that the similarities or differences that one finds for arctic and continental mixed-phase clouds would actually be very interesting for both modellers and observationalists. Therefore, I can only recommend publication of this manuscript when the discussion of former literature related to this study is properly included and discussed.

Pristine Particles: During recent years I realized that people mean different things when they talk about pristine particles. I would like to know more exactly what you mean when you talk about pristine particles. Just single ice particles? Is a single dendrite with some rimed droplets or a broken dendrite still pristine according to your definition? If one reads articles from the in-situ community (e.g. Korolev et al., GRL, 1999) they find even for Arctic clouds in a wide temperature range from 0 to -45°C that only 3% of the observed particles can be classified as being pristine. Considering that one

can expect the conditions for pristine particle growth to be much better in Arctic clouds than in mid-latitude clouds, I wonder how you can be so sure that your remote sensing observations are really related to pristine particles and not to for example polycrystals?

Fall velocity: Throughout the paper you use the term "fall velocity" while I suppose you actually show and talk about the measured radar Doppler velocity. I think you should more carefully distinguish between "terminal fall velocity", "vertical air velocity" and "vertical Doppler velocity". The latter one needs to be first corrected for vertical air motion which is not trivial in ice clouds and I could not find that you applied such a correction. For example, in L. 235 you use "vertical velocity" and actually mean the vertical Doppler velocity. Please be more accurate with these terms throughout the text and the Figure labeling. I am also not sure whether the vertical air motion in mixed-phase clouds is really equally distributed in such a way that long-time averaging of the Doppler velocity would results in the terminal fall velocity of the particles. Do you have any indication for this, maybe from other studies? Because I think this is the basic implicit assumption you are doing here.

Specific comments:

L. 171: I would add to rain attenuation also cloud liquid water in general. Typical attenuation values at Ka-band for cloud liquid water are around (depending on temperature) 1.5 (dB/km) / (g/m$^3$) (see for example Hogan et al., JTECH, 2005). In your case where the liquid water is at cloud top this should not be a big issue but one certainly has to account for it when looking for example at multi-layer mixed-phase clouds like your first example in Fig. 3. Also if the mixed-cloud is at larger heights and the atmosphere is humid, water vapor attenuation cannot be completely neglected at Ka band. In the next sentence you say "strong attenuation is avoided" but what do you consider as "strong"? In fact, any attenuation will introduce a bias in your IWC estimate and hence has to be discussed as a potential source of error. You also state that the 3 dB calibration uncertainty transfers into 30% uncertainty in IWC. However, given the power law in Hogan et al., 2006 for IWC and Z, the IWC error depends on the value of Z. The IWC error for

-10 dBZ due to 3dB uncertainty is certainly larger compared to 0 dBZ.

L. 173: "Radar calibration is estimated to be accurate to 3dB for the LACROS cloud radar". A 3 dB absolute calibration accuracy for a cloud radar certainly needs comprehensive calibration efforts (e.g. external target calibration, long-term calibration monitoring, etc.) which have to go beyond standard manufacturer calibration. I know that programs like ARM invest a lot of money and man power to reach a 3 dB calibration accuracy for their radars so I and probably many readers would be curious to know more details about the calibration efforts you performed to reach this level of radar calibration accuracy.

L. 179-184: From Fig. 6 I find the majority of points being above -23 dB SNR. I can only see that the SNR in general decreases towards higher temperatures. Maybe I missed it, but can you explain this behavior? It is quite unfortunate to use almost the same color (grey) for the data points of LDR between 0 and -14 dB and for the points were a reliable LDR estimate is not possible. Please change.

L. 187: I think I understand what you try to say with your definition of your Z-threshold but for the broader audience I think you should explain a bit more: Why 5000, why -45 dBZ, etc.

L. 215: I can't see why aggregation is the main reason for the low LDRs. Aircraft in-situ data often show that the particles show simply irregular shapes. Also the tendency of the particles to form aggregates in general decreases with lower temperatures.

L. 243: I think this statement is a bit oversimplifying since it is only true if all particles within the volume are perfect Rayleigh scatterers. Particularly close to the dendritic growth region you can hardly be sure that no aggregates are present. Also, if the relation between particle mass and reflectivity would be so straight forward as your statement seems to imply then Z-IWC relations for Ka-band wouldn't show such large errors.

L. 261-268: I see the problem of using the MWR LWP for thin clouds. However, I wonder how well mixed-phase clouds can be approximated with your adiabatic approach given that especially at the top of mixed-phase clouds a lot of mixing due to radiative cooling is taking place and therefore entrainment processes might introduce deviations from the adiabatic LWP estimate. Did you proof that the adiabatic approach is superior to the MWR by plotting for example a scatter plot of MWR LWP vs. adiabatic LWP? For the upper part of your LWP distribution with LWP up to 100-200 g/m$^2$ the MWR estimates should be quite reliable. Are the 20 g/m$^2$ uncertainty referring to the relative or absolute LWP uncertainty? I would assume the relative uncertainties of LWP should be smaller?

L. 278: When you say "strongest peak" does this mean you also consider multimodal spectra? Or do you rather mean "maximum of the radar Doppler spectrum"? I think a statement like "actually makes sense" is not very precise nor scientific, please rephrase. Overall, I do not understand why you are not taking full use of the radar Doppler spectrum itself? You can easily derive IWC for each spectral bin, multiply with the Doppler velocity of the bin and finally integrate the resulting flux. At least you should provide a proof that simply multiplying the moments is a similar good approximation.

L. 291-294: One of your colleagues at IfT just published a study (Kalesse et al., ACP, 2016) that shows the importance to analyze processes and probably also fluxes not along straight vertical profiles but along the fall streaks which can also been seen in your data (e.g. Fig. 3). Have you tried such an attempt?

L. 297: You leave the reader quite alone with this statement. Is this value realistic? Is it consistent with former studies?

Style and Typos:

L. 19-21: I would remove the "and" between "aspects" and "involved" and put another comma instead.

L. 21-22: "Laboratory measurements have already delivered a lot of useful information" sounds quite vague to me. Please be more specific.

L. 23-25: Please provide references to the reader by whom this has been measured.

L. 47: In an "only if" construction it should probably be "is there a direct". Please check.

L. 69-71: Again, please provide the references. Especially in the introduction you should help the reader to find the work you are referring to.

L. 81: "such clouds and may"

L. 94: Please also provide a reference for the radiometer.

L. 179: "an SNR" -> "a SNR"

Caption of Fig. 6: Description is insufficient. Specify what are the single points, the black dots and the error bars.

L. 196: There is no red line in Fig. 7d

L. 230: What do you mean with "raw values"?

L. 208: I think you mean "cloud thickness" here?

L. 255: were investigated

Caption Fig. 10: "averaged for"

L. 279: Comma after "however" missing; "the order of magnitude" of what? The entire sentence is not very clear to me and should be rephrased.

L. 348: Comma before "respectively" missing
* * *

---

## Author Comment (AC1) · 17 Jun 2016

**First Revision of**

"Relation between ice and liquid water mass in mixed-phase cloud layers measured with Cloudnet" by Johannes Bühl, Alexander Myagkov, Patric Seifert and Albert Ansmann

We thank the three reviewers for their detailed comments about our work. The comments and the corresponding action taken are listed below.

According to the reviewers' comments and internal discussions we also made some general changes to the paper:

- The title was changed to: "Measuring ice and liquid water properties in mixedphase cloud layers at the Leipzig Cloudnet station"
- LDR colorscale in plots was changed. Cases for which no LDR could be measured are now shown with empty symbols ("no sig.").
- An error in the programming code for Figure 11 b was corrected and the Figure was exchanged accordingly, decreasing values of the cloud lifetime index by approximately one order of magnitude.

**Anonymous Referee #1**

Overall summary: This manuscript used measurements collected by Leipzig Aerosol and Cloud Remote Observations System (LACROS), which includes Raman lidar, ceilometer, cloud radar and microwave radiometer, and then were analyzed with Cloudnet algorithms to take a detailed insight into the microphysics of mixed-phase cloud layers. Authors found that shallow mixed-phase cloud layers mainly produce pristine ice and spaceborne cloud radar might miss a large part of ice formation. This work presents valuable information to understanding of ice formation and to accuracy of satellite measurements. Some minor questions/suggestions need to be solved are listed in the following:

Comment and Question:

1. Line 64, 97 and 115: Authors should

define the acronyms (TROPOS, LDR, COSMO-EU) when it firstly appeared in the article.

TROPOS: added to Affiliations LDR and COSMO explained at first occurrence

2. A suggestion: the paragraph 2 in page 2 (Line 39-line56) is better moved to the ending of the introduction.

We agree and move this section to the end of the introduction. The following paragraph (Line 59 – line 70 were moved to the front where it makes more sense).

3. As we know, multilayered cloud systems very frequently occur in the atmosphere (Huang, J., P. Minnis, B. Lin, Y. Yi, S. Sun-Mack, T. Fan, and J. Ayers, Determination of ice water path in ice-over-water cloud systems us ing combined MODIS and AMSR-E

measurements, Geophysical Research Letters, 33 (21) (2006), L21801, doi:10.1029/2006GL027038 ; Huang, J., P. Minnis, B. Lin, Y. Yi, M. Khaiyer, R. Arduini, A. Fan, and G. Mace, Advanced retrievals of multilayered cloud properties using multispectral measurements, Journal of Geophysical Research, 110 (D15) (2005), D15S18, doi:10.1029/2004JD005101.). For this study, all lidar or Radar profiles were used? How did you considered the attenuation of lidar power when signal penetrate the lower cloud layer in the multilayered cloud system? There is no correction of lidar attenuation, because the backscatter information from the lidar is not used quantitatively. The lidar is only used for detection of the liquid cloud-top layer. Cloud cases for which the lidar could not (for what reason soever) detect a cloudtop layer for less than 85% of the cloud total occurrence time are omitted due to selection criterion explained in Line 157: "...and at least 85% of the cloud's occurrence time a liquid or mixed-phase cloud top must be detected."

**4. For figure1:**

'On top the predominantly liquid water top is detected by lidar'. As we know that lidar signal is hard to penetrate mix-phase cloud layer, how could it detects the liquid water top? And the schemes of mixed-phase cloud layer are not well described. If there is mixed-phase cloud, why IWP is only below cloud layer?

In mixed-phase cloud layers ice is usually formed at the very top of the layers. However, the freshly formed ice particles are not yet large enough to be detected by lidar or cloud radar. The lidar signal is strongly dominated by the liquid droplets. We changed the sentence above to "Water droplets within the mixed-phase top layer are detected by lidar." hoping to make that issue more clear.

The ice particles fall through the liquid-water-dominated cloud top layer, grow and can soon be detected by cloud radar, but only when they have left the cloud top they can be analyzed quantitatively by cloud radar. Within the mixed-phase cloud layer, signals of cloud droplets and ice crystals cannot be distinguished easily. This is why we analyze the ice particles in the moment when they leave the cloud top layer. When the falling particles are then "alone" the disambiguation between drizzle and ice particles is easier and – as mentioned above – there the properties of the ice crystals can be analyzed quantitatively.

This is why the phase-classification algorithm only take into account particles detected directly below the mixed-phase cloud top layer.

**5. Line 325: 'a minimum cloud layer lifetime of 3 hours around' how could authors get this value**

herein.

The lifetime of 3 hours relates to the value of the "Lifetime index" at -25°C in Fig. 10. In the original manuscript, unfortunately a wrong version of the figure was presented. The correct version of the figure was added to this revised version of the manuscript.

6. For the retrieval of LWC, the adiabatic model was used in this study. However, the entrainment of cloud top should be considered, thus this process may reduce the LWC. A better method possible is based on the depolarization ratio measurement from lidar (See paper: Hu, Y., S. Rodier, K. Xu, W. Sun, J. Huang, B. Lin, P. Zhai, and D. Josset, Occurrence, liquid water content, and fraction of supercooled water clouds from combined CALIOP/IIR/MODIS measurements, Journal of Geophysical Research, 115

(2010), D00H34, doi:10.1029/2009JD012384. Hu, Y., M. Vaughan, C. McClain, M. Behrenfeld, S. Sun-Mack, D. Flittner, J. Huang, B. Wielicki, P. Minnis, C. Trepte, and R. Kuehn, 2007: Global statistics of liquid water content and effective number density of water clouds over ocean derived from combined CALIPSO and MODIS measurements, Atmos. Chem. Phys., 7, 4065-4083.)

Thank You for this interesting hint. For this work, we used LWC values of a purely adiabatic approach. The next step would be to use an advanced method that takes into account measurements of different instruments (e.g., like the paper mentioned). Cloudnet actually delivers LWC measurements of a so-called "scaled adiabatic approach", which scales the LWP measured by a microwave radiometer to the geometrical extent of the cloud. However, our HATPRO microwave radiometer has a bias of 20g/m2 for LWP. Therefore we cannot yet use this method here. Advanced calibration methods are underway (e.g., Maschwitz et. al., AMT, 2013) but unfortunately this is still work in progress.

We added comments about the adiabaticity factor and the alternative methods to the text in Line 280.

**Anonymous Referee #2**

The authors of the manuscript used the Cloudnet algorithm to analyze mixed phase clouds observed in Leipzig. I find the topic interesting and the article is in general worth to be published. However, I have some major comments which should be addressed. In particular, the authors have to address uncertainties more carefully and summarize their findings better so that they not oversell their results.

**1 Major comments**

Abtract: In the abstract, the sensitivity of space borne radars is discussed. However, this issue is mentioned only in Figure 7 and the last(!) sentence of the summary. When mentioning this in the abstract, the reader expects a much deeper discussion of that issue. How is this topic related to the key questions of the paper? I think the authors should either discuss that topic in greater detail in the paper or remove it from the abstract.

We agree with this statement, the discussion of space radar is not sufficiently represented in the paper to be emphasized in the abstract. Hence, it is removed from the abstract.

Nevertheless, we consider the finding interesting and leave it in the paper. We added a sentence in Line 247, discussing the issue more balanced, noting additionally that CloudSat and EarthCare are able to detect most of the ice formation in mixed-phase cloud layers below CTT=-10°C. (A fact that has already been demonstrated in Bühl et. al. GRL 2013 and Zhang et. al.,, JGR, 2010a).

Section 4.1: My major concern is that Cloudnet's IWC estimations are used uncritically. Even though an uncertainty estimate is presented in L 167, the IWC results are likely biased and the authors should make that clear. How does that impact the authors'

results and conclusions? What happens if other Z-IWC relations are used? From the spread of results when using other Z-IWC relations, is it possible to say something about the robustness of the key results of the study?

We consider the Hogan retrieval as the best that is currently available. A comparison with other IWC retrievals is actually envisioned but at the moment we leave that to another study.

We added comments to the text to Section 4.1, explaining the statistical uncertainties and biases given by Hogan (2006).

Moreover, the authors should discus in more detail why is there this drop around -10 °C? How can the authors distinguish between impact of particle type and number concentration?

Based on these measurements this is very difficult to decide. DeMott 2010/2015 show that the number of ice nucleating particles is increasing by a factor of 10 each 5 degree, which would be in the order of magnitude of the IWC increase. However, we think we have no basis to raise this issue in the paper and would rather avoid speculation about this topic.

**Can the authors exclude the possibility of cloud misclassification (i.e. the clouds are actually liquid)?**

Assessing possible misclassifications is difficult, because there is no "truth" dataset or other observations of cloud ice for the clouds observed over Leipzig. We can only compare the results of the algorithm with those of a human observer: The number of mixed-phase clouds found in 5K-temperature intervals between -40 and 0°C are within the errors of the (manual) study of *Bühl et. al, GRL, 2013*. Based on this study the classification accuracy of the algorithm is about 15% in an absolute measure.

**Why is there no significant decrease for very cold temperatures when I would expect smaller particles as well?**

We cannot fully explain this based on our measurements, but since our paper relies on the Hogan retrieval, which actually becomes more precise when going to lower temperatures, we trust into the IWC measurements.

From Fig. 9d, we see that particles indeed become smaller at very low temperatures. Assuming Hogan et. al. 2006 is right, a strong increase in number concentration of particles would explain the issue.

Section 4.2: I cannot see that Figure 8 shows the 'necessity to select thin clouds'. I see that IWC is higher for lower temperatures, but how do the authors now that this is due to more non-pristine particles? I would actually recommend to omit Figure 8 (or explain better why its interesting), because the authors motivated the use of a maximum thickness of 350 m already well with references to other studies.

We agree. In the current context of the paper, Figure 8 is misleading. We leave out the complete Section and move the explanation of thresholds to the introduction.

Section 4.3: What is the influence of vertical air motion? Did the authors correct for that?

Vertical air motion could not be corrected. Correction of vertical velocity has actually been done using a combination of cloud-radar and wind-profiler, (see Bühl et. al., AMT, 2015), but such a system is not available for our study.

We added a comment to the caption of Fig. 9 and to Line 85: "(cloud-radar Doppler velocity average over a complete cloud case)".

Section 4.3: One of the key conclusions of the authors is that 'ice crystals formed in cloud layers with a geometrical thickness of less than 350m are mostly pristine when they fall out of the cloud' which they support by agreement of observations to literature studies. I think this is not supported by the study (except maybe the 0 to -10 °C range). Even though they show indications for the presence of pristine particles, but they show not evidence that other particles are not present. How would non-pristine particles appear in the data with respect to LDR and fall velocity?

Non-pristine crystals that result, e.g., from ice particle break up, aggregation or graupel formation would be asymmetric and would therefore increase the LDR values. However, the LDR values we find are very close to the literature values of *Reinking and Matrosov (1997)*, which yield -28dB for plate-like crystals and about -20dB for columnar shaped particles. These calculations however depend strongly on the orientation of the ice crystals (i.e. at what angle they "wobble"). Larger angles of orientation would increase the measured LDR values. Finding these very low values of down to -30dB is therefore an indication that pristine particles are dominating and secondary ice formation only plays a minor role within the selected cloud layer type.

More precise estimations of this phenomenon can be found in the study of Myagkov (2015) where a ZDR radar technique was applied and particles were also found pristine (including estimations of the angle of orientation.).

We also changed the statement, now claiming that the "dominating part" of the ice crystals found below these clouds is pristine.

Section 4.4: How do the authors know that the median of LWP was around 20 gm-2? From the radiometer of from the adiabatic profiles? Are there differences in the statistic? Even though the general uncertainty of LWP retrievals is around 20 gm-2, the uncertainty is actually way less if environmental conditions (mainly water vapor) are known. As far as I know, LWP estimation is not standardized in Cloudnet. But if LWP is bias corrected before the cloud (i.e constrain the water vapor), I expect the radiometer to be a better source for LWP than the adiabatic profiles. The adiabatic profiles are more an upper threshold for LWP. Moreover, your analysis of ILCR depends on uncertainties and biases of both LWC and IWC.

It is true that the LWP retrieval is not standardized within Cloudnet. We apply a post processing algorithm with background correction which is in an early development phase and not properly evaluated. This algorithm also requires access to the raw data of the radiometer. In this work, we want to develop a general method applicable to any dataset (as stated in Line 68) and the only standardized algorithms of LWP within Cloudnet is the adiabatic method. We also think that this is not sufficient and are looking with great confidence to methods recently developed by other groups (e.g. U. Löhnert, University of Collogne and B. Pospichal, Institut für Meteorologie, Leipzig). But even for these methods, it currently still unclear if they will be able to provide LWP error below about 5 g/m2 which would be needed to detect 20g/m2 and accuracy of 25%. Given

these unsolved issues we think that – for the moment – it is best to proceed with the adiabatic method.

We consider ILCR an estimate, which we try to explain in more detail in Section 4.3. The uncertainty is about one order of magnitude given the biases of IWC and LWC together. Nevertheless the standard-deviation within a temperature interval of about 5°C is only a factor of 2. This might be partially due to the reason that both the IWC and the LWC retrieval method rely on the same temperature field, which reduces this part of the bias.

Section 4.5: As the authors note by themselves, this analysis is only for static conditions. I would propose that this can be seen from the nomenclature as well. I would suggest to call the lifetime index e.g. static lifetime index or potential lifetime index etc. We agree and call the measurement value "static lifetime index".

Summary:

L 303-305: I guess the authors refer to Figure 10? This is interpretation should have been discussed earlier in the manuscript. Moreover, it does not hold given the uncertainties of both LWP and IWC and that the authors show no quantitative evidence that IWC can be really estimated from LWP and CTT.

We agree that the idea of estimating IWC from LWP or vice versa is not developed enough and has no real foundation. It is therefore left out.

L 315: See comment to section 4.3

We changed to "dominating fraction of the ice crystals" (now line 337)

L 340-342: See above, I did not find any relation established and evaluated in this manuscript.

This statement was also left out.

L 346-349: See also comment to abstract: In general, authors should avoid raising totally new issues in the summary as it is done here. Maybe, this part can be moved to the discussion of Figure 7? Is that actually a new, relevant result that pristine ice crystals have low backscattering cross sections and might be missed if the radar sensitivity is not sufficient?

We shifted the discussion about the radar sensitivities to Section 4.2.

2 Minor comments

Title: I find the title of the article misleading. Even though a ratio between LWC and IWC is introduced, it is not discussed in detail.

Perhaps we went to fast with our statement about a "relation".

We changed the title to "Measuring ice and liquid water properties in mixed-phase cloud layers at the Leipzig Cloudnet station", in order to make it more precise.

Figure 6 and other: If the authors plot the lines on top of the dots, the figure would be much easier to read.

We changed the layout of the figures to make them easier to read.

Figure 7a and other: What is indicated by the white boxes?

The white boxes indicate "Maximum values in each column are marked with white bars" which is now mentioned in the figure captions of Figs. 7 and 8.

L. 93: Add Hatpro Reference Added Rose (2005) L. 98: Because the analysis depends on model temperature, what is the impact of this change on your analysis? Is there a bias between the models?

In *Seifert et. al., JGR, 2010* the bias of the model temperatures compared with local radiosonde launches were found to be about 1°K. This error is insignificant compared to the systematic errors of the Hogan retrieval and/or radar calibration.

**L. 118: add 'in Cloudnet below 0**

Sentence changed to "on the basis of temperature only, there is no way to unambiguously decide between drizzle and/or falling ice crystals below 0°C."

L. 120: I found this part hard to understand. Please motivate more clearly why you need this algorithm on top of Cloudnet and what is does

Cloudnet only uses single range gates for ice-particle target classification. We go a step further and use distinct cloud layers in order to decide between mixed-phase and liquid-only clouds. This explanation has been added to Line 139.

L. 124-128: What is the motivation for these thresholds?

Explanation added to Line 147: The 300s horizontal separation is derived from experience. Increasing the value increases quality of the cloud cases. The 350m cloud thickness is motivated by Fukuta(1999), as it probably excludes secondary ice formation processes and particle riming.

L. 125: What is meant by 'driving microphysics'?

Exchanged "driving microphysics" by "cloud properties"

L. 139: I found it puzzling that CB refers to the cloud base of the mixed phase cloud and not of the complete cloud. What is actually the author's definition of a cloud? Are pristine ice particles already precipitation and not considered a cloud anymore? The same holds for Figure 2: 'signal below cloud layer' makes little sense if it is not clear that the mixed cloud layer is meant.

In this work, we rely on the definitions in Fig. 1. The mixed-phase cloud layer has a defined base and top. Particles falling from it are defined as falling (precipitating) particles forming the cloud virga. In Figure 2 "Signal below cloud layer" was exchanged to "virga present?" to make this more coherent.

L. 151: At a typical radar range, what Ze value does -10 dB SNR correspond to?

At 5000m distance the detection threshold of the MIRA-35 Radar is about -45 dBZ. At typical ranges the threshold for LDR detection is -30dBZ. We changed the sentence accordingly.

L. 160: Please motivate thresholds already here.

See above at L. 124-128

L. 187: Is this equation valid for all radars? Where is IWC\_thr in the plot? Caption of Fig. 7 has been changed accordingly

L. 192: Replace dramatically by e.g. significantly done

L. 194: The threshold is black instead of red. In general, such descriptions should be indicated in the caption instead of the text.

corrected

Figure 1: My printer did not print the red ice part of the cloud which makes the figure very hard to understand.

Obviously a problem with PDF export  $\square$  changed to bitmap.

Figure 4: Mention why other clouds are not considered.

Added explanation to figure caption: Clouds are marked due to the selection criteria explained in the text

L 222: Replace 'one has to keep in mind' with 'note' done

L 224: Are the particles investigated here within this interval?

We left out the comment with the Reynolds number because it is misleading. L 250: Isn't the fall velocity increased because these particles are more compact? Yes, this is true. We added "(more compact)" to the text.

L 273: ')' missing done

L 275: Remove actually

done

L 278: too colloquial: 'actually makes sense'

Changed to "can be applied".

L 302: replace 'seem to be' with 'apparently'. 'seems to be' indicates something that is not true in reality

Thank You for this explanation, we changed the text it accordingly.

**Anonymous Referee #3**

The authors present an analysis of a long-term dataset of mid-latitude mixed-phase cloud properties observed by a combination of Ka-band cloud radar, lidar and microwave radiometer. The Cloudnet categorization is used in combination with additional algorithms to estimate liquid and ice water content and ice particle motion to derive ice mass flux.

In general, I find the study interesting and certainly worth for publication in ACP. Unfortunately, the manuscript does not adequately discuss previous work in this field. I also had the feeling that many statements and conclusions given miss proper discussion. I therefore recommend publication after the comments and corrections listed below are addressed.

General comments:

Lack of references and discussion of previous work: This manuscript puts a "special focus on mixed-phase cloud layers" and aims at characterizing heterogeneous ice formation within them with a combination of ground-based remote sensors. However, while reading through the manuscript, I found the given references and discussion of the results of this study with former work to be rather insufficient. Mixed-phase clouds either in the arctic or in mid-latitudes have been a focus topic during the last decade of many institutions and observing programs (for example the ARM program). Nevertheless, I hardly could find any citation of the important work which has been done in this field in this manuscript. I especially miss a proper citation in the introduction but also in the discussion of the results. In my opinion, a proper discussion of the work which has done in this field would also help to put your results into perspective and would actually strengthen your study. Just to give one example: I did just a half an hour literature review on this topic and found for example a very similar comparison between LWP

and cloud top temperature as you show it in Fig. 10a in a recent paper by Zhang et al., "Ice Concentration Retrieval in Stratiform Mixed-Phase Clouds Using Cloud Radar Reflectivity Measurements and 1D Ice Growth Model Simulations", JAS, 2014 (their Fig. 7). Although their paper is about arctic mixed-phase clouds, such work should be discussed in your paper. In fact, I could imagine that the similarities or differences that one finds for arctic and continental mixed-phase clouds would actually be very interesting for both modellers and observationalists. Therefore, I can only recommend publication of this manuscript when the discussion of former literature related to this study is properly included and discussed.

We added several citations about recent ARM activities in the Arctic focused on mixedphase clouds. Also, we included the reference to Zhang et. al. (2014) in line 285. Larson et. al. (2006) and Noh et. al. (2013) were added to Section 4.3.

Pristine Particles: During recent years I realized that people mean different things when they talk about pristine particles. I would like to know more exactly what you mean when you talk about pristine particles. Just single ice particles? Is a single dendrite with some rimed droplets or a broken dendrite still pristine according to your definition? If one reads articles from the in-situ community (e.g. Korolev et al., GRL, 1999) they find even for Arctic clouds in a wide temperature range from 0 to -45 °C that only 3% of the observed particles can be classified as being pristine. Considering that one can expect the conditions for pristine particle growth to be much better in Arctic clouds than in mid-latitude clouds, I wonder how you can be so sure that your remote sensing observations are really related to pristine particles and not to for example polycrystals? With our definition of "pristine" we follow the laboratory measurements of Fukuta (1999). We consider all particles pristine that have not undergone riming growth, aggregation or splintering.

By restricting the extent of the mixed-phase layers to be less than 350m, particles have growth times less than about 20min.

We added to the introduction that particles are considered pristine if they "do not show signs of riming growth or secondary ice formation".

Fall velocity: Throughout the paper you use the term "fall velocity" while I suppose you actually show and talk about the measured radar Doppler velocity. I think you should more carefully distinguish between "terminal fall velocity", "vertical air velocity" and "vertical Doppler velocity". The latter one needs to be first corrected for vertical air motion which is not trivial in ice clouds and I could not find that you applied such a correction. For example, in L. 235 you use "vertical velocity" and actually mean the vertical Doppler velocity. Please be more accurate with these terms throughout the text and the Figure labeling. I am also not sure whether the vertical air motion in mixedphase clouds is really equally distributed in such a way that long-time averaging of the Doppler velocity would results in the terminal fall velocity of the particles. Do you have any indication for this, maybe from other studies? Because I think this is the basic implicit assumption you are doing here.

We added a definition in the introduction at the first occurrence of "vertical velocity", which is defined as cloud-radar Doppler velocity without correction for vertical air motion in Line 85. This makes "fall velocities" the velocity of particles relative to the ground.

The velocity measurements in Figure 8 are averaged over a complete cloud case (minimum 15 minutes), eliminating vertical air motion to a large degree. A general trend is visible showing minimum fall velocities around -17°C with larger fall velocities at -25°C and -5°C. However, a scatter of about +/-0.2m/s is still visible, probably introduced by vertical air motions.

Specific comments:

L. 171: I would add to rain attenuation also cloud liquid water in general. Typical attenuation values at Ka-band for cloud liquid water are around (depending on temperature) 1.5 (dB/km) / (g/m) (see for example Hogan et al., JTECH, 2005). In your case where the liquid water is at cloud top this should not be a big issue but one certainly has to account for it when looking for example at multi-layer mixed-phase clouds like your first example in Fig. 3.

For a usual mixed-phase cloud layer a radar attenuation of about 0.15 db/km is estimated, according to the formula above. Compared to the uncertainties in radar calibration this is neglible. Such small attenuations, (as well as gas attenuation) are automatically corrected in Cloudnet.

Also if the mixed-cloud is at larger heights and the atmosphere is humid, water vapor attenuation cannot be completely neglected at Ka band. In the next sentence you say "strong attenuation is avoided" but what do you consider as "strong"? In fact, any attenuation will introduce a bias in your IWC estimate and hence has to be discussed as a potential source of error.

As stated above, gas attenuation is corrected on the basis of the data from the weather model reanalysis data. This straight forward correction is also usually below 1dB for conditions in Central Europe.

A comment was added to line 202.

You also state that the 3 dB calibration uncertainty transfers into 30% uncertainty in IWC. However, given the power law in Hogan et al., 2006 for IWC and Z, the IWC error depends on the value of Z. The IWC error for -10 dBZ due to 3dB uncertainty is certainly larger compared to 0 dBZ.

Indeed there are small differences for different temperatures and radar reflectivities. We checked again and found a 35% value valid for the range from -40 to 0°C and -60 to 0 dBZ.

A comment was added to line 206.

L. 173: "Radar calibration is estimated to be accurate to 3dB for the LACROS cloud radar". A 3 dB absolute calibration accuracy for a cloud radar certainly needs comprehensive calibration efforts (e.g. external target calibration, long-term calibration monitoring, etc.) which have to go beyond standard manufacturer calibration. I know that programs like ARM invest a lot of money and man power to reach a 3 dB calibration accuracy for their radars so I and probably many readers would be curious to know more details about the calibration efforts you performed to reach this level of radar calibration accuracy.

Radar calibration is done by measuring transmitted power at the radar feed. Since the LACROS radar is equipped with a scanner this procedure can be relatively easily be

accomplished with a large building nearby. The calibration itself is done regularly by METEK company and we rely on their expertise. Full characterization of the MIRA-35 radar is given in Görsdorf et. al. (2015), which we now cite within the paper.

L. 179-184: From Fig. 6 I find the majority of points being above -23 dB SNR. I can only see that the SNR in general decreases towards higher temperatures. Maybe I missed it, but can you explain this behavior? It is quite unfortunate to use almost the same color (grey) for the data points of LDR between 0 and -14 dB and for the points were a reliable LDR estimate is not possible. Please change.

We changed the colors. Point for which no LDR measurement is available are now marked empty (or white). We think that SNR decreases toward lower temperatures due to smaller particle size and larger distance to the targets.

L. 187: I think I understand what you try to say with your definition of your Z-threshold but for the broader audience I think you should explain a bit more: Why 5000, why -45 dBZ, etc.

The explanation of the threshold was changed and formulated more clearly with a new (corrected) formula for the range-dependent Z-threshold.

L. 215: I can't see why aggregation is the main reason for the low LDRs. Aircraft in-situ data often show that the particles show simply irregular shapes. Also the tendency of the particles to form aggregates in general decreases with lower temperatures. The respective section was removed due to suggestions of Reviewer #2.

L. 243: I think this statement is a bit oversimplifying since it is only true if all particles within the volume are perfect Rayleigh scatterers. Particularly close to the dendritic growth region you can hardly be sure that no aggregates are present. Also, if the relation between particle mass and reflectivity would be so straight forward as your statement seems to imply then Z-IWC relations for Ka-band wouldn't show such large errors.

The discussion of Rayleigh/Mie effects would go beyond this study. We decided to omit the respective sentence.

L. 261-268: I see the problem of using the MWR LWP for thin clouds. However, I wonder how well mixed-phase clouds can be approximated with your adiabatic approach given that especially at the top of mixed-phase clouds a lot of mixing due to radiative cooling is taking place and therefore entrainment processes might introduce deviations from the adiabatic LWP estimate. Did you proof that the adiabatic approach is superior to the MWR by plotting for example a scatter plot of MWR LWP vs. adiabatic LWP? For the upper part of your LWP distribution with LWP up to 100-200 g/m the MWR estimates should be quite reliable. Are the 20 g/m uncertainty referring to the relative or absolute LWP uncertainty? I would assume the relative uncertainties of LWP should be smaller?

The HATPRO correction algorithm must correct for several different issues (rain, wet radome, multiple scattering, sensor drift) and is currently optimized for other scenarios. This restriction can lead to negative values of LWP (at maximum about -5g/m2) which can occur under mixed-phase cloud conditions, due to an unknown contribution of water vapor radiation. Since the adiabatic approach only take into account the cloud geometry

it is robust. The error due to the unknown adiabaticity is actually lower than using the microwave radiometer data for these clouds. We also need a robust method, that is applicable to any Cloudnet dataset. As explained above, new correction algorithms are underway but implementation is not yet completed.

L. 278: When you say "strongest peak" does this mean you also consider multimodal spectra? Or do you rather mean "maximum of the radar Doppler spectrum"? I think a statement like "actually makes sense" is not very precise nor scientific, please rephrase. Overall, I do not understand why you are not taking full use of the radar Doppler spectrum itself? You can easily derive IWC for each spectral bin, multiply with the Doppler velocity of the bin and finally integrate the resulting flux. At least you should provide a proof that simply multiplying the moments is a similar good approximation. As we understand the retrieval of Hogan (2006) is not designed in a way that it could be applied to single spectral bins. Also spectra often are broadened due to turbulence and have a positive (upward-moving) component, which results in unphysical results.

L. 291-294: One of your colleagues at IfT just published a study (Kalesse et al., ACP, 2016) that shows the importance to analyze processes and probably also fluxes not along straight vertical profiles but along the fall streaks which can also been seen in your data (e.g. Fig. 3). Have you tried such an attempt?

We actually tried such approaches for case studies but the procedure is automatized in a way that it could be applied automatically to a large dataset. The method of Heike Kalesse is very powerful when multi-layered clouds are analyzed, which we consider the next step. In the current work, we only focus on ice that is falling from the cloud top layer and try to measure it in the moment when it leaves cloud top, so fall-streak tracking is not really applicable. But as noted before, when multi-layered clouds are involved fallstreak tracking is a powerful tool, because it can show the ice evolution and can give information about the influence of different liquid sub layers on the cloud particles. In addition, our approach is based on averages over a full coherent cloud layer which removes the potential need to perform fall streak tracking. This is certainly an interesting subject for future studies.

L. 297: You leave the reader quite alone with this statement. Is this value realistic? Is it consistent with former studies?

As noted in the beginning, we exchanged the figure and changed the sentence in order to make it more clear.

**Style and Typos:**

L. 19-21: I would remove the "and" between "aspects" and "involved" and put another comma instead

done

L. 21-22: "Laboratory measurements have already delivered a lot of useful information" sounds quite vague to me. Please be more specific.

Changed the sentence to: "Laboratory measurements have already delivered a lot of useful information, e.g., about the ice nucleation efficiency of aerosol particles with temperature Murray (2012), DeMott (2015)."

L. 23-25: Please provide references to the reader by whom this has been measured.

We think that the word "flux" appeared too early in the manuscript. We changed to IWC and added references.

L. 47: In an "only if" construction it should probably be "is there a direct". Please check. done

L. 69-71: Again, please provide the references. Especially in the introduction you should help the reader to find the work you are referring to.

References are given. (Now in line 45.)

L. 81: "such clouds and may"

Added the missing text: "Such cloud are difficult to observe and may..."

L. 94: Please also provide a reference for the radiometer.

Added Rose et. al. (2005).

L. 179: "an SNR" -> "a SNR"

done

Caption of Fig. 6: Description is insufficient. Specify what are the single points, the black dots and the error bars.

Description extended.

L. 196: There is no red line in Fig. 7d

"red line removed from the text"

L. 230: What do you mean with "raw values"?

Clarifying statement was added: (30-s integration time and 30 m height resolution)

L. 208: I think you mean "cloud thickness" here?

This part has been removed following comments of Reviewer #2.

L. 255: were investigated

done

Caption Fig. 10: "averaged for"

done

L. 279: Comma after "however" missing; "the order of magnitude" of what? The entire sentence is not very clear to me and should be rephrased.

[revised manuscript text omitted]

---

## Author Response (AR2)

**Reply to 2nd Review of "Relation between ice and liquid water mass in mixed-phase cloud layers measured with Cloudnet" by Johannes Bühl, Patric Seifert, Alexander Myagkov, and Albert Ansmann**

We thank Reviewer#2 very much for scrutinizing the manuscript again. We believe that this made the manuscript considerably better and regret that we did not meet all critical points before. In our response below we try again to correct the errors and clarify the issues raised.

I thank the authors for enhancing the manuscript. However, I still have concerns regarding three of the major issues I raised. Remarkably, the authors rejected in particular comments which involve more effort than changing some lines in the manuscript.

Major comments

Vertical air motion: The authors have to provide more details in L 153/154: How much averaging is required to get rid of vertical air motion? Are there any references supporting that? Isn't that in opposition to their statement in the introduction: "Vertical motions keep mixed-phase clouds alive by activating aerosol particles to cloud droplets…".

Influence of turbulent motion induced by the cloud layer itself averages out during an observation time of 15 minutes. The Doppler velocity shown in Fig. 8b and d is, however, still susceptible to large-scale motions longer than 15 minutes. We believe that a lot of the scatter in Fig. 8b and d is resulting from such influences.

A comment about that issue was added at Line 168.

Why did the authors not apply the Shupe et al 2008 algorithm for detecting vertical air motion? The authors seem to know the method since the article is in their references. At least the authors should mention the issue more clearly when analyzing the Doppler velocity and introducing the static ice flux.

The algorithm of Shupe 2008 separates particle fall velocity from the turbulent properties of a cloud-top layer with the help of a Fourier analysis. In the process, the velocity is "mean-removed" and "detrended". Hence, the method can only provide information about vertical air motions with frequencies smaller than the cloud observation time. It could therefore not be used to remove the influence of large-scale air motion from the average Doppler velocity observed within falling particles. For that, an independent and direct measurement of vertical air motion at cloud top or directly within the falling particles (see Bühl et. al., AMT, 2015) would be needed.

Furthermore, the authors often use the term 'fall velocity' when they actually mean 'Doppler velocity' (e.g. Fig 8b and corresponding discussion). Please go through the manuscript in order to use them correctly.

We agree and changed "fall velocity" to "Doppler velocity" wherever we talk about measurements.

Z-IWC relation: The authors responded: "We consider the Hogan retrieval as the best that is currently available." but do not point out why they think Hogan's method is best. Hogan et al (2006) assumed the mass-size relation of Brown and Francis (1995) which is based on a study by Locatelli and Hobbs (1974). They developed the mass-size relation for "aggregates of unrimed bullets, columns and side-planes". Since this study is targeting pristine ice particles, how can a mass size relation for aggregates be used? Moreover, Hogan assumed the median volume diameter of the particle size distribution to be between

0.2 and 3 mm. Is that in agreement with the expected size of the pristine particles? I would recommend to apply 2-3 other Z-IWC relations in order to investigate whether this changes the overall results. I expect this won't change the results, but give a better feeling for the uncertainties.

We believe that the temperature dependence actually implicitly takes into account particles shape in the retrieval. The relationships have been elaborately validated by Heymsfield, JAMC, 2007 (DOI: 10.1175/2007JAMC1606.1) and Protat, JAMC, 2007 (DOI: 10.1175/JAM2488.1). In the following graph we retrieved the Ice Water Content on the basis of the values of CTT and the mean values of Z shown in Figure 7b in the paper (thick lines). We compare 4 different Z-T parameterizations of Hogan and Protat. We conclude that the spread is about a factor of two for the temperature interval -30 to -15°C and about a factor of 5 in the interval -15 to 0°C.

We also now give this information in the paper (Line 207).

[Figure]

LWP retrieval: I still think that a radiometer retrieval would be better suited to get LWP if it is offset corrected during clear sky periods. This would correct not only for water vapor, but also for potential calibration offsets. Since a radiometer is part of the standard Cloudnet set up I do not see why this should limit the applicability of the approach to other data sets. Did the authors actually compare LWP values of the radiometer and the adiabatic approach to each other?

We compared the values and the results are plausible in a statistical sense. However, the unknown bias for an individual measurement is about 20g/m². For individual cases, we found that the background correction algorithm sometimes created implausible negative values of LWP or values that are much too high. The current retrieval would therefore only introduce larger uncertainties and strongly increase the scatter of the measurements. Also the negative LWP measurements would introduce a dangerous bias. The adiabatic retrieval has the geometrical cloud thickness and the adiabaticity factor as an uncertainty, but usually yields plausible values.

Minor comments

Cloud misclassification: The authors responded that they estimate the accuracy of their cloud

classification to be around 15% (I hope that this is actually the inaccuracy!). I would suggest to mention this in the manuscript.

We mention this now in Line 198.

Pristine particles: I would suggest that the authors point out in section 4.2 more clearly why the data shows that the observed particles are pristine. In their response to my corresponding comment, the authors points this out more clearly than in section 4.2.

We added a description, trying to clarify the usage of the term "pristine" starting from Line 293.

Static lifetime index: 'static' is not used consistently in the manuscript. e.g. Section 4.4 and Figure 10.

Replaced

- "static approaches" by "approaches" (Line 306)
- "Steady conditions" → "static conditions" (Line 359)
- "Static lifetime parameter index" → "static lifetime index" (Line 389)

L 147: I do not understand this statement, how is 'quality' quantified?

In this case, we actually talk about the "homogeneity" of the cloud cases. By demanding a larger separation between the clouds, the number of cloud cases will go down, but the clouds will be less influenced, e.g. from gravity waves originating from other clouds nearby.

We added this explanation to the manuscript (Line 148).

L204: This calibration is 1) component based (i.e. theoretical) and 2) for a different radar.

2: The manufacturer of the radar (METEK company) does the calibration method described in Görsdorf and Bauer (2015) in the same way for all of its systems.

1: This calibration is indeed component based, so some unknowns remain. An advantage of the LACROS radar is, e.g., that the antenna is characterized experimentally because the radar is equipped with a scanner unit.

We try to explain this better in the manuscript at Line 220.

L 221: I'm surprised that the authors use such a static formula. Don't the authors use the standard MIRA moments which finds the hydrometeor peak in the Doppler spectrum?

The threshold formula only relates to the sensitivity of the detector. In principle, it gives the radar constant.

L255 fall velocities are decreased in 8b, but enhanced in 8d for the highest temperature bin?

The increase at the highest temperature bin is an interesting feature because it is also predicted by the measurements of Fukuta and Takahashi. However, in our measurements the significance of this increase is not large enough.

We added an explanation to the text: "Actually, the increase of Doppler velocities in the temperature interval between -5 and 0°C bin is also found in Fukuta (1999). However, the uncertainty of the measurements in this temperature interval is too large for a definite identification of this phenomenon."

Figure 8: Is it possible that different temperature bins have been used in Figure 8a and c (b and d)?

All temperature bins are 2.5°C and we indeed found that they were slightly shifted (by about 1K). We recalculated the values with the correct intervals, but significant changes in the mean values are not visible. There is also a basic difference between the two values: The white bars indicate the maxima of the histograms for each temperature bin and the lower ones show the mean values.

L 292/293: A 'future' approach published in 2010? Please rephrase.

Replaced "future" by "alternative"

L 298: If the IWC uncertainty is one magnitude, how can the uncertainty of IWC/LWC be one magnitude as well?

We agree that this formulation is wrong. We try to explain the errors of the corresponding measurements in more detail in Lines 329.

L 295 & 304: Isn't the statement in line 295 "Assuming that particles directly below the mixed-phase layer have the same properties as within the layer…" (i.e. the authors assume static conditions) in opposition to Line 304 "ice formation is a dynamic process"?

We agree that the term "dynamic process" creates some misunderstanding at this point and leave it out.

L 301: Do the authors really mean 'error'? maybe 'variability'?

We changed to "variability".

L 308: v -> v_CB?

Indeed, v and v_CB have been used in the manuscript inconsistently. We updated the manuscript accordingly.

Autors' response to reviewer #3: I noticed that the authors mentioned a minimum time span for a cloud case of 15 min, but according to the manuscript (L 155) it is 20 minutes.

I'm very sorry for this mistake: The selection criterion actually is 15 minutes. It has been mentioned in the manuscript several times, but one time wrongly. We checked the manuscript again for this error.

**Other changes:**

Fig. 9: "ILCR ratio" → "ILCR"

Line 307 → added "water content" to definition of ILCR